

# A model for the spatial distribution of snow water equivalent parameterised from the spatial variability of precipitation

Thomas Skaugen[1] and Ingunn H. Weltzien[1,2] *

[1] {Norwegian Water Resources and energy Directorate, P.O. Box 5091, Maj. 0301 Oslo, Norway.}
[2] {Department of Geosciences, University of Oslo}

Correspondence to: T. Skaugen (ths@nve.no)
*now at Norconsult AS, P.O. Box 626, 1303,Sandvika Norway



## Abstract

Snow is an important and complicated element in hydrological modelling. The traditional catchment hydrological model with its many free calibration parameters, also in snow sub-models, is not a well-suited tool for predicting conditions for which it has not been calibrated. Such conditions include prediction in ungauged basins and assessing hydrological effects of climate change. In this study, a new model for the spatial distribution of snow water equivalent (SWE), parameterized solely from observed spatial variability of precipitation (SD_G), is compared with the current snow distribution model used in the operational flood forecasting models in Norway. The latter model (SD_LN) has a fixed, calibrated coefficient of variation, which parameterizes a log-normal model for snow distribution. The two models are implemented in the already parameter parsimonious rainfall runoff model Distance Distribution Dynamics (DDD) and their capability for predicting runoff, SWE and snow covered area (SCA) are tested and compared for 71 Norwegian catchments. Results show that SD_G better simulates SCA when compared with MODIS satellite derived snow cover. In addition, SWE is simulated more realistically in that seasonal snow is melted out and the building up of "snow towers" and giving spurious positive trends in SWE, typical for SD_LN, is prevented. The precision of runoff simulations using SD_G is slightly inferior, with a reduction in Nash-Sutcliffe and Kling Gupta Criterion of 0.01, but it is shown that high precision in runoff prediction using SD_LN is accompanied with erroneous simulations of SWE.

**Key words:** Distribution of snow, SWE, SCA, runoff, hydrological modelling



## 1    Introduction

Snow is an important hydrological parameter in the northern hemisphere and in Norway approximately

30 % of the annual precipitation falls as snow. Snow and snow related hydrology have a significant impact

on nature and society in such regions. Seasonal snow ensures variation in outdoor activities and

considerable investments in infrastructure for tourism and hydropower are dependent on stable seasonal

snow. Apart from snow related hazards such as spring melt floods and avalanches, snow may negatively

affect construction safety and traffic flow at airports, roads and in urban areas. Information of snow

conditions at the local, regional and national scale is therefore important for the early warning of hazards,

but also for tourism, hydropower production planning and water resources management.

Operational snow models have evolved differently for hydrology than for meteorology and avalanche

warning. Whereas the model development in the latter two scientific disciplines usually include detailed,

multi-layered, physically based process representations, snow models in hydrology are typically calibrated

empirical relationships between snow variables and the modest model forcing at hand, i.e. snow

accumulation and melt vs precipitation and  temperature. An example of such a calibrated relationship is

the degree-day model for snowmelt (Hock, 2005; Ohmura, 2000), where snowmelt is a linear function of

the difference between air temperature and a (often calibrated) temperature threshold for which there is

no snowmelt. In practise, the degree-day factor is calibrated against runoff, and will hence account for a

multitude of processes and scales. One reason for such a discrepancy in modelling approaches is that

calibrated hydrological snow models have proved themselves at low temporal resolutions (i.e. 24h

resolution (Anderson, 1976)) and for stationary climatic conditions. Another reason is that hydrological

snow models are expected to provide simulations at the catchment scale, for which there are usually no





estimates of more non-standard hydrological model forcing such as, for example, wind and radiation. In
addition, the governing equations for the physics of hydrology at the small scale have proven difficult to
scale up in time and space to be relevant for catchment hydrology (Kirchner, 2006).
For predictions in ungauged basins and in a changed climate, however, calibrated empirical relations in
snow models cannot be expected to give reliable and useful results. As an example, Skaugen et al. (2015)
used the Distance Distribution Dynamics (DDD) model (Skaugen and Onof, 2014) for predicting in
ungauged basins with model parameters estimated from catchments characteristics. When analysing the
deviations in performance between the calibrated and the regionalised versions of the DDD model, the
regionalised degree-day factor for snowmelt and the coefficient of variation for the spatial distribution of
SWE emerged as the parameters most responsible for poor regionalised results for runoff.
In this study we will investigate how snow water equivalent (SWE), snow covered area (SCA) and runoff
are simulated when an alternative method for parameterising the spatial distribution of SWE is
implemented in a hydrological model. The method has all its parameters estimated prior to calibration and
is described in Skaugen (2007) and has since been developed in Skaugen and Randen (2013). The method
models the spatial probability density function (PDF) of SWE as a dynamic gamma distribution and is
hereafter denoted SD_G (Snow Distribution_Gamma)). SD_G was tested at small test sites and found to
model the spatial moments of SWE and SCA well (Skaugen and Randen, 2013), but has, however, not
been implemented in a hydrological model and hence not been tested for larger scales and as a tool in
operational hydrology. A realistically modelled PDF of SWE is important for the temporal evolution of
SWE, snowmelt and SCA (Luce and Tarboton, 2004; Essery and Pomeroy, 2004; Luce et al., 1999; Liston,
1999; Buttle and McDonnel, 1987). Good simulation of the evolution of SCA is especially important since




it controls the runoff dynamics of the spring melt flood and is the basis for properly accounting the energy
fluxes in land- surface schemes in atmospheric models (Helbig et al., 2015; Essery and Pomeroy, 2004;
Liston, 1999). In addition, remotely sensed SCA is one of the few data measured at the catchment scale
for which simulated hydrology can be compared, and represents hence a valuable independent data source
to validate models.
The parameters of SD_G are estimated solely from observed spatial variability of precipitation. Such
information is available at many sites which makes it possible to use the method for prediction in ungauged
basins. Downscaled climate changes projections may also provide such information so that effects of
climate change on snow conditions and hydrology  may be assessed. In using such a method, the current
dependency of calibration in  hydrological snow models is reduced.
It is not straightforward to evaluate new process algorithms in, sometimes, heavily parameterized rainfall-
runoff models.  Due to the tendency for calibration parameters to compensate for data- and structural
model errors (Kirchner, 2006; Beven and Binley, 1992), it can be difficult to identify the effect of changing
an algorithm or parameterization from the calibrated results of the model (Clarke, 2011a). Kirchner (2006)
points out the need to develop models that are minimally parameterized, and which therefore stand a
chance of failing the tests they are subjected to, which is exactly the problem faced when assessing the
introduction of new algorithms with fewer calibration parameters. Gupta et al. (2014) propose the use of
large sample hydrology as a means for the testing of hypothesis and model structures, in order to a) arrive
at conclusions more general than can be achieved using a single catchment, b) establish a range of
applicability, and c) ensure sufficient information to enable the identification of statistically significant
relationships.  In addition, a minimal use of calibration parameters should increase the efficiency in





isolating and demonstrating the effect of new algorithms and parameterizations (Kirchner, 2006).
Consequently, in this study large sample hydrology *and* a parameter parsimonious model are used to
investigate the suitability of a new model algorithm.
We will implement the SD_G model in the parameter parsimonious DDD rainfall-runoff model. In DDD's
current snow routine the spatial PDF of SWE is modelled as the sum of uniform- and log-normally
distributed snowfall events (Sælthun 1996, Killingtveit and Sælthun, 1995). The distribution is constant
for up to a specified threshold of accumulated SWE (i.e. 20 mm). Each additional snowfall event is log-
normally distributed through a calibrated coefficient of variation (CV) and SWE is estimated for nine
quantiles and added to previous quantile values. In this way, each additional snowfall event has a spatial
distribution of a fixed shape (through the calibrated CV) regardless of its intensity. Moreover, the method
implies perfect spatial correlation in that a new snowfall is distributed such that the quantiles with highest
SWE always receives most SWE so that the CV of the sum of snowfall events remains a constant. A
simple example demonstrates this: if the accumulation of snow $Z$, is the sum of two snowfall events $y$,
$Z = y_1 + y_2$ where $y \sim LN(\mu_y, \sigma_y^2)$ is log-normally distributed, then the mean of $Z$ is $E(Z) = 2\mu_y$ and the
variance is $Var(Z) = \sigma_y^2 + \sigma_y^2 + 2COV(y_1, y_2)$. With perfect correlation the variance equals $Var(Z) =$
$\sigma_y^2 + \sigma_y^2 + 2\sigma_y^2$ (Haan, 1977, p.56) and it is easily seen that the that the coefficient of variation for $Z$
equals that of $y$, i.e

$$CV_Z = \frac{\sigma_Z}{\mu_Z} = \frac{2\sigma_y}{2\mu_y} = CV_y$$

The spatial distribution of melt is constant and reduction in SCA occurs when the SWE associated with a
quantile becomes zero. The fraction of snow-free areas is thus the sum of quantiles with zero SWE. This



snow distribution model is hereafter denoted SD_LN (Snow Distribution _Log-Normal) and, although
SD_LN has been used operationally in Norwegian hydrology for many years, it has the feature of being a
calibrated model and hence not suitable for climate change studies and for predictions in ungauged basins.
In addition, an assumption of perfect spatial correlation and hence a fixed CV is not supported by
observations. Timeseries of spatial measurements of SWE in Norway show that the spatial CV vary
through the snow season (Alfnes et al., 2004; Skaugen, 2007). In Skaugen and Randen (2013) SD_LN
was found inferior to SD_G in years for which it was not calibrated and recently Frey and Holzmann
(2015) published a study that shows that a log-normal spatial distribution of SWE with a fixed CV of
introduced so called "snow towers". For high elevation areas, and for the highest quantiles of the
distribution, snow survived the summer and accumulated to give an overall positive trend in SWE which
was not observed.
The main objective of this paper is to evaluate if a method for describing spatial PDF of SWE with
parameters estimated a priori calibration is a suitable alternative for use in rainfall runoff models. We will
compare simulated results of SWE, runoff and SCA simulated with DDD using the current model, SD_LN
and with the alternative, SD_G for 71 catchments in Norway. Time series of satellite-derived SCA
(MODIS, Moderate Resolution Imaging Spectroradiometer) is available for the catchments, so simulated
runoff and SCA will also be compared against observed values.
**2      Method**





The proposed method requires that we represent the spatial PDF of SWE by an analytical model. In the
literature, many such models are proposed, especially for the period of time of maximum accumulation;
such as the log-normal (Donald et al. 1995, Sælthun 1996), the gamma (Kutchment and Gelfan, 1996;
Skaugen, 2007; Kolberg and Gottschalk, 2010; Skaugen and Randen, 2013) and the normal (Marchand
and Killingtveit, 2004, 2005). Helbig et al. (2015) investigated the spatial PDF of snow depth for three
large alpine areas and found that the gamma and the normal distributions were better suited than the log-
normal. In Alfnes et al. (2004), Skaugen (2007) and in Skaugen and Randen (2013), it was demonstrated
through the repeated measurements of the same snowcourse during the accumulation and melting seasons
that the spatial PDF of SWE changed it shape continuously during the periods of accumulation and
melting. During the accumulation period, the spatial distribution of SWE would become less positively
skewed as accumulation progressed and increasingly more positively skewed as melting progressed. Since
we aim to have an estimate of the spatial PDF of SWE at all times during the snow season, we continue
here the approach outlined in Skaugen (2007) and Skaugen and Randen in (2013), modelling the spatial
PDF of SWE as a sum of gamma distributed correlated unit fields.

## 2.1   Moments of spatial SWE

We need, at all times, estimates of the spatial conditional mean, $E(Z')$ and variance $Var(Z')$, of
accumulated SWE. The PDF of $Z'$ does not contain zeros and is hence conditional on snow. For the non-
conditional distribution of SWE, which also includes zeros, the variable SWE is denoted $Z$. The notation
of $Z$ will hereafter determine if we discuss the conditional or the non-conditional spatial distribution of $Z$.
The spatial conditional PDF of SWE is modelled as a gamma distribution with shape and scale parameters:



$$v = \frac{E(Z')^2}{Var(Z')} \text{ and } \alpha = \frac{E(Z')}{Var(Z')} \tag{1}$$

As in Skaugen and Randen (2013), the PDF of accumulated SWE is approximated by the sum of correlated
gamma distributed unit fields, $y(x)$, where x represents space. For the remainder of this paper the unit
$y(x)$ is denoted $y$. The unit fields of snowfall are distributed in space according to a two-parameter gamma
distribution, $y = G(v_0, \alpha_0)$ with PDF:

$$f(y) = \frac{1}{\Gamma(v_0)} \alpha_0^{v_0} y^{v_0 - 1} e^{-\alpha_0 y}, \quad \alpha_0, v_0, y > 0 \tag{2}$$

where $\alpha_0$ and $v_0$ are parameters . The mean of the unit equals $E(y) = v_0/\alpha_0$ and the variance equals
$Var(y) = v_0/\alpha_0^2$. When estimating the moments for the sum of $n$ units, $Z'(n) = \sum_{i=1}^{n} y_i$ we have to take
into account that the unit fields are correlated. This has no bearing on the mean, $E(Z')$ but affects the
variance, $Var(Z')$, i.e:

$$E(Z') = n \frac{v_0}{\alpha_0} = \frac{v}{\alpha} \tag{3}$$

$$Var(Z') = n \frac{v_0}{\alpha_0^2} + 2 \sum_{i<j} Cov(y_i, y_j) = n \frac{v_0}{\alpha_0^2} [1 + (n-1)c(n)] = \frac{v}{\alpha^2} \tag{4}$$

where the function $c(n)$ is the average correlation over $n$ units.
From Eq. (4) we see that if we have perfect and constant correlation between the $y$'s, $c(n) = 1$, the
variance of $Z'$ equals $Var(Z') = n^2 \frac{v_0}{\alpha_0^2}$ and by Eq. (3) we have that the relationship between the standard
deviation, $\sigma_{Z'}$, and the mean, $E(Z')$ is a straight line with the slope equal to $v_0^{-0.5}$, $\sigma_{Z'} = v_0^{-0.5} E(Z')$ . On
the other hand, if we have no correlation between the $y$'s, $c(n) = 0$, the variance equals $Var(Z') = n \frac{v_0}{\alpha_0^2}$
which gives a relationship between $\sigma_{Z'}$ and $E(Z')$ as a curved line that departs from that of perfect



correlation by $n^{-0.5}$, $\sigma_{Z\prime} = (v_0 n)^{-0.5} E(Z')$. The variance, however, is linearly related to the mean.
Correlation between the units, $c(n)$ gives a relationship between the mean and the standard deviation that
is something between the two cases described above. A typical analytical approximation to the spatial and
temporal correlation function for precipitation is an exponentially decaying function with either time or
space as argument. Zawadski, (1973, 1987) found exponential decorrelation for rainfall for both time and
space. As $n$ (number of summations) may be considered a variable akin to time, $c(n)$ is approximated by
an exponential correlation function:

$$c(n) = \exp\left(-\frac{n}{D}\right),$$
(5)

where $D$ is the decorrelation range where the correlation equals $1/e$ (Zawadski, 1973).
The variance of $Z'$ can now, with eqs. (4) and (5), be expressed as:

$$Var(Z') = E(Z') \frac{1}{\alpha_0} [1 + (n-1)exp(-n/D)]$$
(6)

From measured, positive (i.e. not including zeros) precipitation over an area we can observe the
relationship between the spatial mean and spatial variance of precipitation. Furthermore, we can estimate
the two unknowns, $D$ and $\alpha_0$ from such data by nonlinear regression. Figure 1 a) shows a scatterplot of
spatial mean and standard deviation of positive precipitation with a fitted function of the type Eq. (6).
From Figure 1 b), where the spatial man and standard deviation are plotted in a log-log space, we see that
the relationship is not that of a power law, as suggested in Skaugen and Randen (2013) and Skaugen and
Andersen (2010), since a straight line does not represent the point cloud very well.





During the snow season, the snowpack may experience a series of melting and accumulation events and
estimating the temporal variability of the spatial variance of SWE is clearly a challenge. Furthermore,
SCA varies throughout the season, which necessarily adds to this complexity. In this study SCA is set
equal to 1 (full coverage) for every snowfall event, whereas a melting event implies a reduction in
coverage. In the following subsections we will briefly address the estimation of the mean and variance of
SWE for accumulation and melting events under different conditions of snow coverage. The derivation
for accumulation events differs from that presented in Skaugen and Randen (2013) and is presented in
detail. For melting events and for the estimation changes in SCA, however, only the resulting equations
are presented since the full derivations can be found in Skaugen and Randen (2013).

## 2.2    Moments of spatial SWE after an accumulation event

From a single snowfall event of $n$ units on a snow-free surface, the mean and the variance of the snow
reservoir $Z'$ are estimated according to eqs. (3) and (4). The parameters $a_0$, $v_0$ and $D$ are estimated from
a priori analysis of the local variability of precipitation (see Figure 1). A mean of the units has been chosen
as $E(y) = \frac{v_0}{a_0} = 0.1 \, mm$, since $0.1 \, mm$ is the smallest precipitation value measured by the Norwegian
Meteorological Institute. If there is an additional snowfall event of $u$ units, the mean and the variance of
the resulting snow reservoir are simply:
The mean:

$$E(Z'_{n+u}) = (n + u)\frac{a_0}{v_0} \tag{7}$$



and the variance:

$$Var(Z'_{n+u}) = \frac{v}{\alpha^2} + u\frac{v_0}{\alpha_0^2}[1 + (u-1)c(u)], \tag{8}$$

where $\frac{v}{\alpha^2}$ is the conditional variance prior to the accumulation event. In order to keep the notation simple
we say that $n$ is the number of units at $t-1$ and $u$ is the number of units of the event at time $t$.
Equations (7) and (8) are valid if $SCA = 1$ for both events. If SCA prior to the second event has been
reduced due to melting ($SCA_{t-1} < 1$), we have to scale the contributions of $n$ and $u$ according to the
change in SCA from, $SCA_{t-1} < 1$ to , $SCA_t = 1$, hence:
The mean

$$E(Z'_{n+u}) = \frac{a_0}{v_0}(SCA_{t-1}(n+u) + SCA_t u) \tag{9}$$

And the variance

$$Var(Z'_{n+u}) = SCA_{t-1}^2(\frac{v}{\alpha^2} + u\frac{v_0}{\alpha_0^2}([1+(u-1)c(u)])) +$$

$$SCA_t^2\frac{v_0}{\alpha_0^2}u([1+(u-1)c(u)]) \tag{10}$$

## 2.3    Melting events

15    Let the snow reservoir, consisting of $n$ units, be reduced by $u$ units after a melting event. The snow

16    coverage before and after the melting event is $SCA_{t-1}$ and $SCA_t$ respectively, where $SCA_t < SCA_{t-1}$. We

17    set $SCA_{t-1}$ as 1, so that $SCA_t$ is the relative reduction in snow coverage due to melting, and not the



catchment value. Reduction in snow coverage needs special attention regarding the conditional ($Z'$) and
the non-conditional ($Z$) moments since we have to determine the spatial moments for the area of the new
coverage $SCA_t$ (not including zeros, i.e. conditional moments) and for the area which includes the
previously covered part, $SCA_{t-1}$ (including zeros, i.e non-conditional moments).

### 2.3.1  The spatial mean after a melting event

The non-conditional mean after the melting event, is estimated as:

$$E(Z_{n-u}) = (n-u)\frac{v_0}{\alpha_0}$$
(11)

and the conditional mean is

$$E(Z'_{n-u}) = \frac{E(Z_{n-u})}{SCA_t} = \frac{1}{SCA_t}(n-u)\frac{v_0}{\alpha_0}$$
(12)

We note that the difference in conditional means before and after the melting event is

$$E(Z'_n) - E(Z'_{n-u}) = \frac{v_0}{\alpha_0}\left(n-(n-u)\frac{1}{SCA_t}\right) = \frac{v_0}{\alpha_0}(u')$$
(13)

where $u'$ is the conditional number of melted units and describes the difference in units when the (relative)
reduction in SCA is taken into account.

### 2.3.2  The spatial variance after a melting event





Skaugen and Randen (2013) give a detailed derivation of the conditional spatial variance of SWE after a
melting event. Here, only the final expression is reported:

$$Var(Z'_{n-u}) = \frac{v}{\alpha^2} - 2u'n\frac{v_0}{\alpha_0^2}c_{mlt}(u') + u'\frac{v_0}{\alpha_0^2} + u'(u'-1)\frac{v_0}{\alpha_0^2}c(u') \qquad (14)$$

where $\frac{v}{\alpha^2}$ is the variance of $Z'$ prior to the melting event, and $c_{mlt}(u')$ is the (negative) correlation between
melt and SWE and is estimated as a linearly decreasing function of $u'$ and equal to:

$$c_{mlt}(u') = \frac{u'}{n}\left(\frac{1}{2n}\left(\frac{v}{\alpha^2}\frac{\alpha_0^2}{nv_0}+1+(n-1)c(n)\right)\right), \qquad (15)$$

It is clear from Eq. (13) that an estimation of the change in SCA due to melting, which will be presented
in the next subsection, is needed in order to assess $u'$ and consequently $Var(Z'_{n-u})$ in Eq. (14).

## 2.4    Estimating changes in snow covered area (SCA)

After a snowfall event, the SCA for the area of interest (a catchment or a part of a catchment in the case
of elevation bands) is set equal to 1. For a melting event, however, the estimate of changes in SCA is
more complex. The previous subsections suggest modelling the accumulated SWE as a gamma
distribution, $f_a$, with parameters $v$ and $\alpha$ derived from the estimated mean and variance as described above.
In Skaugen and Randen (2013), also the spatial frequency of snowmelt, $f_s$ was modelled as a gamma
distribution, following the same the same principles as for accumulation, i.e that the moments can be
estimated using eqs. (3) and (4) with $u'$ replacing $n$. At all times $u' \le n$, which implies that until the final
melting event occurs, $f_s$ is more skewed to the left than $f_a$. The correlation of snowmelt $c(u')$ as a function





of intensity ($u'$) has not yet been investigated in detail and is, in this study, modelled as that of
accumulation.  Skaugen and Randen (2013) however, reported empirical evidence supporting such an
assumption with the respect for the features of $f_s$, i.e. that the spatial distribution is generally skewed to
the left and becomes less skewed as the intensity of melt increases. These features are confirmed by
additional measurements of spatial snowmelt by Weltzien (2015).
Figure 2 illustrates how the reduction in SCA due to a melting event is estimated. Since the energy
requirements for transforming a snowpack into snowmelt is linearly related to snow depth (Dingman,
2002), it is reasonable to assume that areas with the smallest values of SWE are the first to become snow
free, i.e. we assume a perfect (negative) correlation between SWE and snowmelt. Since $f_a$ and $f_s$ are
*spatial* frequency distributions of SWE and snowmelt respectively, the frequencies can be interpreted as
number of locations and their integral as fractions of an area. In Figure 2, the value $X$ defines the value of
SWE/snowmelt where the frequencies of the melt distribution, $f_s$, are higher or equal to the frequencies
of the accumulation distribution, $f_a$. All locations with SWE values less than the value $X$ are hence left
snow-free which constitutes a fractional area of $\int_0^X f_a = a$. When the frequencies (number of locations)
of $f_a$ are higher than those of $f_s$, only a fraction of these locations will be snow-free. The sum of these
fractions amounts to $\int_X^\infty f_s = 1 - s$, (see Figure 2). The total reduction in SCA after a melting event is
thus:

$$SCA_{red} = a + 1 - s \qquad (17)$$

Recall that the reduction in SCA is relative, i.e. it is the reduction from the previous snow-cover which is
also the probability space of both $f_a$ and $f_s$, and equal to 1.



**2.5. The hydrological model**
The DDD model (Skaugen and Onof, 2014; Skaugen et al., 2015; Skaugen and Mengistu, 2015) is a
rainfall runoff model written in the programming language **R** (www.r-project.org) and runs operationally
at daily and 3-hourly time steps at the Norwegian flood forecasting service at the Norwegian Water
resources and Energy Directorate (NVE). The DDD model introduces new concepts in its description of
the subsurface and of runoff dynamics and is developed with the objective of having as many as possible
of its model parameters estimated prior to calibration from  observed data such as maps and runoff
characteristics. In its current version, the parameters of the modules for subsurface- and runoff dynamics
are all estimated prior to calibration against runoff. Estimating parameters of the subsurface from
estimated mean celerity and observed mean annual runoff is a new development and is described in
Skaugen and Mengistu (2015). Input to the DDD model is precipitation and temperature. The model is
semi-distributed in that the moisture-accounting (rainfall and the accumulating and melting of snow) is
performed for ten elevation bands of equal area. The catchment averages of precipitation and temperature
are distributed to the elevation bands using calibrated lapse rates. Snowmelt is estimated using a degree-
day model (Hock, 2005; Ohmura, 2000) where the melted amount is a linear function of the difference
between actual air temperature and a calibrated threshold temperature for melting. The catchment
averaged precipitation can be corrected in order to get the long-term water balance right.



The model parameters relevant for snow accumulation and melt which are estimated by calibration against
runoff include $\theta_{CV}$, which describes the spatial distribution of SWE, $\theta_{CX}$ which is the degree- day factor
and $\theta_{WS}$, which is the maximum liquid water content in the snowpack (see Table 1 of model parameters).
Further details on the DDD model is found in the cited literature. Model parameters calibrated against
runoff are hereafter denoted by $\theta$ with subscripts (e.g. $\theta_{CV}$), in order to clearly distinguish between
estimated and calibrated parameters. From Table 1 we see that altogether 11 model parameters can be
calibrated.

## 9 2.5 Test of SD_G in DDD

We will evaluate the performance of SD_G, parameterised from from observed spatial variability of
precipitation, implementing it in DDD (DDD_G) and compare performance with DDD_LN, in which
SD_LN , with its calibration parameter $\theta_{CV}$, is implemented.
The new parametrization of the subsurface is tested for 71 catchments distributed across Norway (see
Figure 3). The catchments vary in latitude, size, elevation and surface cover (see histograms of selected
catchment characteristics in Figure 4) and constitute thus a varied, representative sample of Norwegian
catchments.
The following procedure was followed: the models were initially calibrated using long time series of
precipitation and temperature to simulate runoff using a Monte-Carlo Markov-Chain method (Soetart and
Petzhold, 2010) written in **R**. The time series for precipitation and temperature are mean areal catchment





values extracted from the current, operational meteorological grid (1 x 1 km²) which provides daily values
of precipitation and temperature for Norway from 1957 to the present day (see www.senorge.no).  This
meteorological grid is denoted V1.  Recently, a new improved meteorological grid was developed, denoted
V2, (Lussana et al. 2014a, Lussana et al. 2014b) which reduced much of the positive bias in precipitation
characteristic of V1 (see Saloranta, 2012). The new meteorological grid (V2) in DDD gives reasonable
simulated values of runoff without the need for a calibrated correction of the amount of precipitation ($\theta_{Pc}$,
see Table 1 for parameters of the DDD model). Areal averages of precipitation and temperature values are
extracted for ten elevation zones which makes it possible to eliminate calibrated precipitation and
temperature gradients ($\theta_{Plr}$ and $\theta_{Tlr}$). Three parameters associated with snow accumulation and melt (the
correction factor for solid precipitation ($\theta_{Sc} = 1.0$), the threshold temperature for snowmelt ($\theta_{Ts} = 0\ °C$)
and the threshold temperature for solid and liquid precipitation ($\theta_{TX} = 0.5\ °C$) were fixed, thereby
reducing the number of calibration parameters from 11 to 5. For the remaining 4 parameters, the calibrated
values (from using V1 as input) are retained for 3 parameters ($\theta_{Ws}$, $\theta_{v_r}$, and $\theta_{cea}$), whereas for the
DDD_LN model, $\theta_{CX}$ and the parameter of interest for this study $\theta_{CV}$, is recalibrated using V2 as input
data.  In using such a procedure we assume that the 3 parameters which are calibrated using the V1 data
(and, most likely, not optimal for the V2 data as input) will not favor either of the two compared model
structures (calibrated (SD_LN)- and estimated (SD_G) spatial distribution of SWE). When recalibrating
the $\theta_{CV}$ with V2 data, we attempt to make it as difficult as possible to accept the new spatial frequency
distribution of SWE. If we calibrated all 3 parameters ($\theta_{Ws}$, $\theta_{v_r}$, and $\theta_{cea}$)  using V2, we could risk that
errors associated with the structures of SD_G and SD_LN were compensated for by the other 3 parameters,
such that we could not isolate and evaluate the effect of implementing SD_G. Also for the DDD_ G model,





the degree-day factor $\theta_{CX}$, was calibrated since correlation between this parameter and $\theta_{CV}$ was revealed.
It would hence be probable that a $\theta_{CX}$ optimised using SD_LN with V1 would not be optimal for testing
SD_G.
With the procedure described above, we can compare the performances of the DDD model with calibrated
PDF of SWE (DDD_LN ) and the DDD model with estimated PDF of SWE (DDD_G) with respect to
runoff, SWE and SCA.
**3    Results**
**3.1 Runoff**
Figure 5 shows different skill scores obtained for runoff simulations for the 71 catchments with DDD_LN
(red crosses) and DDD_G (blue circles). ). Figure 5 a) shows the Nash-Sutcliffe efficiency criterion (NSE,
Nash and Sutcliffe, 1970) and 5 b) the Kling-Gupta Efficiency criterion (KGE, Gupta, et al. 2009, Kling
et al. 2012)   and 5 c-e) the three components of the KGE, correlation, bias and variability error,
respectively. The variability error is given by the ratio of the coefficients of variation of simulated and
observed runoff as suggested in Kling et al. (2012). The mean values of the skill scores for DDD_LN and
DDD_G are shown as straight lines in the plots and in Table 2. We see from the Figure 5 and Table 2 that
little precision in predicting runoff is lost when using DDD_G. The mean values for NSE, KGE, and the
different elements of KGE are practically identical.
**3.2    SWE**



Figure 6 shows an example of a timeseries of simulated SWE using DDD_G (blue) and DDD_LN (red).
This example illustrates what was seen for most catchments with reliable occurrence of seasonal snow.
SWE simulated with DDD_LN tends to survive the summers at the highest elevations, which results in a
positive trend for SWE. Seasonal SWE simulated by DDD_G and DDD_LN is similar at the start of the
time series but deviates increasingly as time proceeds.    Figure 7 a) shows a scatterplot of the mean
simulated SWE (averaged over the timeseries) for the 71 catchments by the two models and it is clearly
seen that SWE simulated by DDD_LN is higher than simulated by DDD_G although both precipitation
and temperature are identical for the two models. From linear regression between SWE, precipitation and
temperature with time we can estimate simple annual trends. Figure  7 b, c, d)  shows plots of the slopes
of the regression lines. Whereas both precipitation and temperature show very modest annual rates of
change, both models simulate increasing SWE with time, but DDD_LN, on average, 5 times as much as
DDD_G.  If we estimate that 100 days of solid precipitation represent the average accumulated SWE, the
increase in SWE due to the positive trend in precipitation comes very close to the trend in SWE found for
DDD_G.
**3.3 Snow covered area**
From almost 1500 optical satellite scenes from MODIS during the period 2001- 2015, SCA for each
elevation band have been estimated for the 71 catchments. Many scenes are discarded due to insufficient
light caused by the low solar angle during the Norwegian winter, but for each catchment, about 150
estimates of SCA during the 15 years can be used for validation of the snow distribution models' ability
to simulate a realistic evolution of snow free areas during the snowmelt period. For each MODIS satellite
scene, each pixel (500 X 500 meters) is assigned a SCA value between 0-100% coverage using a method



based on the Norwegian linear reflectance to snow cover algorithm (NLR) (Solberg et al. 2006). The input
to the NLR algorithm is the normalized difference snow index signal (NDSI- signal) (Salomonsen and
Apple, 2004). Figure 8 shows the root mean square error (RMSE) between observed and simulated
catchment values of SCA for 69 catchments (two of the catchments did not have SCA observations).
Although the mean RMSE does not differ much between the two models (mean RMSE = 0.14 for for
DDD_G and mean RMSE = 0.15 for  DDD_LN) we can note that SCA is better estimated using SD_G
for 46 out of 70 catchments (66%). Figure 9 shows a typical example where SD_G has estimates of SCA
close to the observed especially during late spring. Naturally, the problem of "snow towers" of SD_LN
influences its ability to simulate a realistic decrease in SCA since small fractions of the catchments remains
snow covered at all times. We can also note, from Figure 9, that SD_LN appears to have a more realistic
start of the reduction of SCA than SD_G which might be a consequence of that the log-normal distribution
may be quite positively skewed. Such a distribution obviously has a higher frequency of small values of
SWE and hence, give an earlier reduction in SCA.
**4     Discussion**
Table 2 shows that, according to the Nash-Sutcliffe and Gupta-Kling efficiencies, the models are almost
identical with respect to the simulation of runoff. This implies that little performance is lost in simulating
runoff by introducing the new procedure for modelling the spatial frequency distribution of SWE although
there are one parameter less to calibrate.  A reduction in the number of parameters to calibrate reduces the
dimensions of the parameter space and thus the parameter uncertainty. In addition, it makes the model less
flexible and more dependent on its structure so that possible structural deficiencies more easily can be





identified (Kirchner, 2006). These are very important points when the demands on hydrological models
moves from just predicting runoff to reliable predictions for more elements in the hydrological cycle such
as for example SWE and SCA. In addition, to properly assess the hydrological effects of climate change
and to provide useful predictions for ungauged basins, we have to move towards the use of hydrological
models with a minimum of calibration parameters.
An important objective of this study, besides that of reducing the number of calibration parameters, is to
investigate whether DDD_G gives a more realistic simulation of snow properties, such as a realistic
temporal evolution of SWE and SCA. Figures 6 and 7 show that DDD_LN gives clear a positive trend for
simulated SWE, whereas DDD_G gives a small positive trend in SWE that corresponds roughly to that of
precipitation (recall that SWE is the accumulated solid precipitation during a period of time). It is notable
that such an obvious erroneous simulation of SWE using SD_LN has so little impact on the precision of
runoff predictions. A possible reason is that the surplus of snow, located at the highest elevations and for
small areal fractions, will not have temperatures high enough, even during summer, to generate intense
snowmelt affecting the precision of runoff simulations. It is, however, of concern that the method itself
introduces trends that could easily be interpreted as a trend in SWE in a climatic study. This problem of
"snow towers" in models using a log-normal distribution for SWE with a fixed, calibrated CV has recently
been addressed in the literature (Frey and Holzmann, 2015). In Norway, using such a snow distribution
model with the well known Swedish rainfall-runoff model, HBV (Hydrologiska Bråns
Vattenbalansmodell, (Bergström, 1992)) has led to the operational procedure of deleting the remaining
snow reservoir at the end of summer. Such a procedure clearly constitutes an example of a model working
well (with respect to runoff) but not for the right reasons. This point is further illustrated when we focus





on one of the catchments that gives better NSE values using DDD_LN than DDD_G. The Masi catchment
(5543 km$^2$) is located in northern Norway and is relatively flat (90 % of its area is located below 600
m.a.s.l and its minimum and maximum elevation is 370 and 1085 m.a.s.l respectively) so that the snow
melt season is quite short and intense. Figure 10 a) shows the simulation of SWE using SD_LN with
optimised CV (CV= 0.88) which gave a NSE value for runoff of NSE=0.75 and using SD_G which gave
a NSE value for runoff equal to NSE=0.72. In Figure 10 b) we have adjusted the CV value from CV=0.88
to CV=0.1 and the simulation of SWE using SD_LN no longer exhibit the very strong positive trend seen
in Fig. 10 a), looks more realistic and very similar to that of SD_G. The precision of runoff simulation
was, however affected and the NSE value dropped from NSE= 0.75 to NSE= 0.60. A reasonable
conclusion may thus be that the slightly higher values for NSE and KGE using SD_LN is at the expense
of unrealistic values of SWE.
Figure 8 shows that in general, SCA is better simulated using SD_G than SD_LN. From figures 8 and 9
we see that the "snow towers", or heavy tails of the optimised accumulation distribution produced by
SD_LN make a complete melt-out of the snow reservoir very difficult. SD_G, on the other hand, provides
an accumulation distribution without the heavy tail, which appears as a better choice with respect to the
simulation of both SWE and SCA. A more realistically simulated SCA is important for many applications.
In overparameterized rainfall runoff models, the optimal runoff simulation is often a system of
compensating errors in states (i.e. soilmoisture and  SWE) and updating one of  the states from
observations may, in fact, deteriorate the simulation result because the balance of errors is disturbed
(Parajka et al. 2007). Updating of snow- and hydrological models using observed SCA is hence dependent
on realistic simulations of SCA. A realistic simulation of SCA is also necessary for the properly accounting




of energy fluxes over an area partly covered by snow (Liston, 1999; Essery and Pomeroy, 2004) and is
hence important for the assessment of hydrological impacts of climate change. Without realistically
simulated SCA, we cannot expect credible simulations for climate projections for neither runoff dynamics
nor energy fluxes.
SWE is represented here as the sum of correlated (in time) spatial variables (solid precipitation).
Precipitation events as snow is assumed to be gamma distributed in space with parameters varying with
intensity.  The parameters, scale, $\alpha_0$ and decorrelation length $D$ are estimated from observed spatial
moments of precipitation. Recall that the shape parameter $\nu_0$, is just set as one tenth of $\alpha_0$ through the
relation $E(y) = \frac{\nu_0}{\alpha_0} = 0.1\ mm$. From Figure 1 we see that the variance levels off at a certain spatial mean
intensity, and even decreases. The presented model captures this observed feature since the variance will
cease to increase as the correlation decreases with intensity (number of summations). For uncorrelated
events, we finally have a sum of independent events. According to the central limit theorem, such a sum
will have a normal distribution. The shape parameter of $y$, $\nu_0$ and the correlation determines the rate of
the convergence to a normal distribution. For example, if the decorrelation range is long, then more
summations are needed for the spatial frequency distribution of SWE to approach a normal distribution.
The literature shows that empirical spatial distribution of SWE has a tendency to be positively skewed.
This is especially the case for many observations of SWE in Norway in high alpine areas (Alfnes et al.
2004; Marchand and Killingtveit, 2004; Marchand and Killingtveit 2005). For more lowland and forested
areas, the distribution tend to be more normal (Alfnes et al, 2004; Marchand and Killingtveit, 2004;
Marchand and Killingtveit 2005). In our modelling framework, this would imply that we would expect
small shape parameters and long decorrelation lengths for mountain areas, and larger shape parameter



together with short decorrelation lengths for lower lying forested areas. Table 3 show correlations and
their significance (p-values) between the parameters $\alpha_0$ and $D$ with catchment values of fraction of bare
rock, fraction of forest, mean elevation and catchment area.
We see that $\alpha_0$ is significantly correlated to the mountain/forest highland/lowland indices as expected.
The decorrelation length $D$ is weakly correlated to the mean elevation in a way implying shorter correlation
lengths at high altitudes, i.e. contrary to what is expected from reported shapes of the PDF of SWE, and
uncorrelated to the other indices. It is promising, and somewhat unexpected, that correlation between
$\alpha_0(\nu_0)$ and catchment characteristics supports our theory so clearly since the location of Norwegian
precipitation gauges, which is has a very poor representation at high elevations (Dyrrdal et al. 2012;
Saloranta, 2012), was not expected to discriminate this behaviour very well. The somewhat confusing
results of the decorrelation length, suggests a dedicated study using a more dense network of precipitation
gauges.
As mentioned in section two, many models for the spatial distribution of SWE have been proposed in the
literature (i. e. normal, gamma, log-normal, mixed log-normal). The diversity in distributions is often
addressed to the physical processes responsible for the shape of the spatial distribution of SWE, which
include wind, during and after the snowfall, spatial variability of precipitation and topographic features.
This topic is continuously debated in the literature (Scipion et al. 2013; Clark et al., 2011; Mott et al.,
2011; Lehning et al., 2008; Skaugen, 2007; Liston, 2004) and, in addition, various spatial scales and
landscape types interact and further complicate the matter (Liston, 2004; Alfnes et al. 2004; Marchand
and Killingtveit, 2004; Marchand and Killingtveit, 2005; Blöschl, 1999). A major problem is that the
spatial distribution of snow and SWE is very hard to measure at the appropriate scale, i.e. catchment scale,





which often covers different elevations and both forested and open (alpine) areas. Various airborne
observation techniques such as laser scan (Melvold and Skaugen, 2013) and passive microwave
(Vuyovich, 2014) are promising but are restricted by landscape features such as vegetation and topography
and the state of the snow (wet/dry). Consequently, investigations on the spatial distribution of SWE has
to rely on in situ measurements which seldom covers entire catchments. In this study, in situ information
(the spatial variability of solid and liquid precipitation), is obtained from the station network of the
Norwegian Meteorological Service, which measures precipitation at 2 m above ground. It is highly
probable that the observed spatial variability, measured at such near-surface, captures information of the
influence of the wind on precipitation in general and on snowfall in particular. This assumption is justified
by the significant and relatively high correlations seen in Table 3 between the scale parameter (and hence,
in our case, the shape parameter) of the distribution of $y$ to landscape features such as elevation and
vegetation and suggests a sensitivity to the exposure of wind. Johansson and Chen (2003) demonstrate the
influence of wind speed on the spatial distribution of precipitation and Mott et al. (2011) and Lehning et
al. (2008) show that near-surface wind fields highly influence snow distribution patterns through
preferential deposition.
The method presented in this study does not include redistribution of SWE due to wind as a driving force
shaping the spatial frequency distribution of SWE at the catchment scale. Some authors suggest that this
process occur on a spatial scale much smaller than the catchment scale (Melvold and Skaugen, 2013;
Liston, 2004). In Figure 9 we see that SD_LN show a better simulation of SCA for the start of the melting
period than SD_G for, at least, two of the years. The reason to why SD_LN simulates the initial
development of snow-free areas better  than SD_G is probably that SD_LN is generally more positively





skewed than SD_G, and has, hence, a higher frequency of small values of SWE that melts quickly.
Whereas the distribution of SD_G, in general, seem to be more appropriate, a fraction of the catchment
should perhaps be populated with small values of SWE in order to simulate this observed initial
development of snow-free areas. By including redistribution due to wind, we might produce areas of
shallow SWE, such as over wind exposed ridges which are known to become free of snow rather early in
spring.
Finally, it is important to keep in mind that this study aims at determining the spatial frequency distribution
of SWE for elevation bands for a catchment. These areas may comprise several square kilometres. The
spatial distribution of SWE for distributed hydrological modelling, i.e. simulating the amount of SWE at
specific locations, is another, and much more challenging, task which involves taking into account very
small scale (< 25 m according to Lehning et al. 2008) landscape features and their complex relation to
accumulation, melting and redistribution of SWE.
**5      Conclusions**
In this paper a method for estimating the spatial frequency distribution of  SWE is implemented in the
parameter parsimonious rainfall- runoff model DDD. The new method, first described by Skaugen (2007)
and further developed by Skaugen and Randen (2013), has its parameters estimated from observed spatial
variability of precipitation measured from precipitation gauges.  The new method (SD_G) has hence no
parameters to be optimized from calibration against runoff unlike the current operational snow distribution
routine (SD_LN), which has one calibration parameter. The new method gives a dynamic presentation of
the distribution of SWE, which, at the start of the accumulation season may be positively skewed, but





converges to a more symmetrical distribution as the accumulation season progresses. The parameters of
the method show significant correlations with catchment characteristics discriminating between sheltered
and wind exposed areas.
SD_G is tested for 71 catchment in Norway and little loss in precision of predicted runoff is seen when
skill is measured with the Nash-Sutcliffe and Kling-Gupta efficiency criteria. SWE is simulated more
realistically in that the seasonal snow is melted out every year and no trend in SWE is observed, which is
consistent with the absence of trends in precipitation and temperature. The current operational routine for
snow distribution (SD_LN), however, displays a tendency to produce ever increasing "snow towers" (Frey
and Holzmann, 2015), which in turn gives the erroneous impression of an increasing trend in SWE. Such
a behaviour can be remedied by adjusting the optimised CV value but at the expense of the precision of
simulated runoff. The simulated SCA for both SD_G and SD_LN is compared to MODIS derived SCA
and SD_G has the lower RMSE. The difference in simulated SCA between the two models is especially
seen for median to low values of SCA. SD_LN can be sees to simulate better SCA at the beginning of the
melt season, suggesting that SD_G has a too low frequency of low SWE values.
**Acknowlegdement**
The help of Nils Kristian Orthe at NVE in providing the satellite derived SCA data is gratefully
acknowledged. This work was partly conducted in the project "Better SNOW models for natural hazards
and HydropOWer applications- SNOWHOW " (project 244153) funded by the Norwegian Research
Council. NVE supports an open data policy and real-time, and near real-time data is available at



http://www.nve.no/en/Water/Data-databaser/Real-time-hydrological-data/ and historical data is freely
available at request to hydrology@nve.no.





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

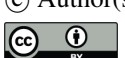


**Table1.** Parameters of the DDD model with comments and method of estimation. Some parameters (denoted with a *) have
fixed values obtained through experience in calibrating DDD for gauged catchments in Norway. These values are within the
recommended range for the HBV model (Sælthun,1996). Other parameter values are assigned standard values as suggested in
the literature. The GIS analyses are carried out using the national 25 X 25 m DEM (www. statkart.no). Parameters in bold are
calibrated.

| Parameter | Comment | Method of est. | Value | Ref |
|---|---|---|---|---|
| Hypsograpic curve | 11 values describing the quantiles 0,10,20,30,40,50,60,70,80,90,100 | GIS | | |
| $\boldsymbol{\theta_{Ws}}$ [%] | Max liquid water content in snow | Calibrated (V1) | 5 | |
| Hfelt | Mean elevation of cathment | GIS | | |
| $\theta_{Tlr}$ [°C/100 m] | Temperature lapse rate for (pr 100 m) | Standard value | 0.0 | |
| $\theta_{Plr}$ [mm/100 m] | Precipitation gradient (mm per 100 m) | Standard value | 0.0 | |
| $\theta_{Pc}$ | Correction factor for precipitation | Standard value | 1.0 | |
| $\theta_{Sc}$ | Correction factor for precipitation as snow | Standard value | 1.0 | |
| $\theta_{TX}$ [°C] | Threshold temperature rain /snow | Standard value | 0.5 | |
| $\theta_{TS}$ [°C] | Threshold temperature melting / freezing | Standard value | 0.0 | |
| $\boldsymbol{\theta_{CX}}$ [mm/°C/day] | Degree-day factor for melting snow | Calibrated (V1) | | |
| $C_{Glac}$ [mm/°C/day] | Degree-day factor for melting glacier Ice | * | 1.5x$\boldsymbol{\theta_{CX}}$ | Sælthun (1996) |
| $CFR$ [mm/°C/day] | Degree-day factor for refreezing | * | 0.02 | Sælthun (1996) |
| Area[m$^2$] | Catchment area | GIS | | |
| maxLbog[m] | Max of distance distribution for bogs | GIS | | |
| midLbog[m] | Mean of distance distribution for bogs | GIS | | |
| Bogfrac | Fraction of bogs in catchment | GIS | | |
| Zsoil | Areal fraction of zero distance to the river network for soils | GIS | | |
| Zbog | Areal fraction of zero distance to the river network for bogs | GIS | | |
| $NOL$ | Number of storage levels | Standard value | 5 | Skaugen and Onof (2014) |



| $\boldsymbol{\theta_{cea}}$ [mm/°C/day] | Degree day factor for evapotranspiration | Calibrated (V1) | | |
|---|---|---|---|---|
| $R$ | Ratio defining field capacity | Standard value | 0.3 | Skaugen and Onof (2014) |
| $\delta$ | Shape parameter of gamma distributed recession characteristic | Estimated from recession | | |
| $\beta$ | Scale parameter of gamma distributed recession characteristic | Estimated from recession | | |
| $\boldsymbol{\theta_{CV}}$ | Coeff. of variation for spatial distribution of snow | Calibrated (V2) | | |
| $\alpha_0$ | Scale parameter of unit precipitation | Estimated from observed spatial variability of precipitation | | |
| $D$ | Decorrelation length of spatial precipitation | Estimated from observed spatial variability of precipitation | | |
| $\boldsymbol{\theta_{v_r}}$ [m/s] | Mean celerity in river. | Calibrated from (V1) | | |
| $m_{Rd}$[m] | Mean of distance distribution of the river network | GIS | | |
| $s_{Rd}$[m] | Standard deviation of distance distribution of the river network | GIS | | |
| $Rd_{max}$[m] | Max of distance distribution in river network | GIS | | |
| $m_S$[mm] | Mean of subsurface water reservoir | Estimated from recession | | |
| $\bar{d}$[m] | Mean of distance distribution for hillslope | GIS | | |
| $d_{max}$[m] | Max of distance distribution for hillslope | GIS | | |
| Glacfrac | Fraction of bogs in catchment | GIS | | |
| $m_{Gl}$[m] | Mean of distance distribution for glaciers | GIS | | |
| $s_{Gl}$[m] | Standard deviation of distance distribution for glaciers | GIS | | |
| Areal fraction of glaciers in elevation zones | 10 values | GIS | | |





**Table 2** .Mean values of skill scores obtain with simulating with DDD_G and DDD_LN for 71 catchments. KGE_r measures
correlation, KGE_b, the bias error and KGE_g the variability error. All skill scores have an ideal value of 1.

|        | NSE  | KGE  | KGE_r | KGE_b | KGE_g |
|--------|------|------|-------|-------|-------|
| DDD_G  | 0.64 | 0.70 | 0.85  | 0.85  | 1.02  |
| DDD_LN | 0.65 | 0.71 | 0.85  | 0.84  | 0.99  |

**Table 3**. Spearman correlations between model parameters and catchment characteristics indicating alpine and lowland areas
where the spatial distribution of SWE is expected to vary. The bracketed numbers indicate significance (p-value)

|              | %Forest     | %Bare rock   | Mean elevation | Catchment size |
|--------------|-------------|--------------|----------------|----------------|
| $\alpha_0$   | 0.34 (0.00) | -0.40 (0.00) | -0.35 (0.00)   | -0.28 (0.02)   |
| $D$          | 0.13 (0.29) | -0.14 (0.24) | -0.25 (0.03)   | -0.15 (0.19)   |





Figure captions
**Figure 1**. Scatter plot of the spatial mean and spatial standard deviation of observed precipitation over a catchment.
Equation (6) is fitted to the data by non-linear regression (red line). Bottom panel shows the scatter plot in log-log.
**Figure 2.** Schematic of how changes in SCA is estimated. $f\_s$ and $f\_a$ are the spatial frequency distributions (PDF)
of snowmelt and accumulation respectively. $s$, $1 - s$, $a$ and $1 - a$ are partial integrated values of the PDFs.
**Figure 3**. Location of the 71 catchments used to evaluate the new subsurface routine
**Figure 4**. Histograms of catchment characteristics for the 71 catchments. a) mean of the hillslope distance
distribution, $\bar{d}$, b) areal percentage of lakes, c) areal percentage of bogs, d) catchment area , e) mean elevation, f)
areal percentage of glaciers, g) areal percentage of forests and h) areal percentage of bare rock.
**Figure 5.** Skill scores for DDD_G (blue circles) and DDD_LN (red crosses) for 71 Norwegian catchments. Mean
skill score values are shown in horizontal lines (same color code).a) NSE, b) KGE, c) KGE_r (correlation), d)
KGE_b (bias) and e) KGE_g (variability error).
**Figure 6**. Time series of simulated SWE using DDD_G (blue line) and DDD_LN (red line) for catchment Tansvatn
in Southern Norway.
**Figure 7.** Scatter plot of mean SWE simulated with DDD_G and DDD_LN for 71 catchments (a), scatterplot of
annual slope of SWE b), annual slope of precipitation c) and temperature d).
**Figure 8.** Root mean square error of SCA for DDD_G (blue) and DDD_LN (red). Moving average of RMSE and
the mean RMSE are shown with the same color code.
**Figure 9.** Time series of simulated SCA with DDD_G (blue) and DDD_LN (red) together with MODIS derived
SCA (green circles) for catchment Tansvatn in Southern Norway.





1    **Figure 10.** Time series of simulated SWE for the Masi catchment in northern Norway with DDD_G (blue) and

2    DDD_LN (red). In a) SWE is simulated with optimised CV=0.77, which gives a NSE=0.75. In b) SWE is simulated

3    with adjusted CV=0.1 which gives a NSE=0.60. Using DDD_G gives a NSE=0.72.





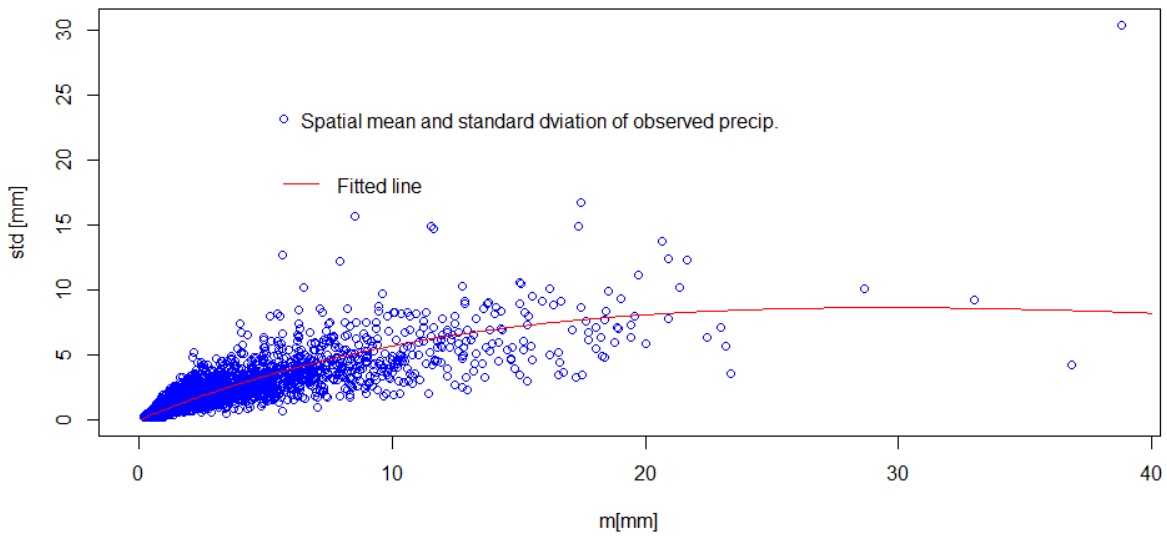

3    Fig 1





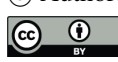



3    Fig 2







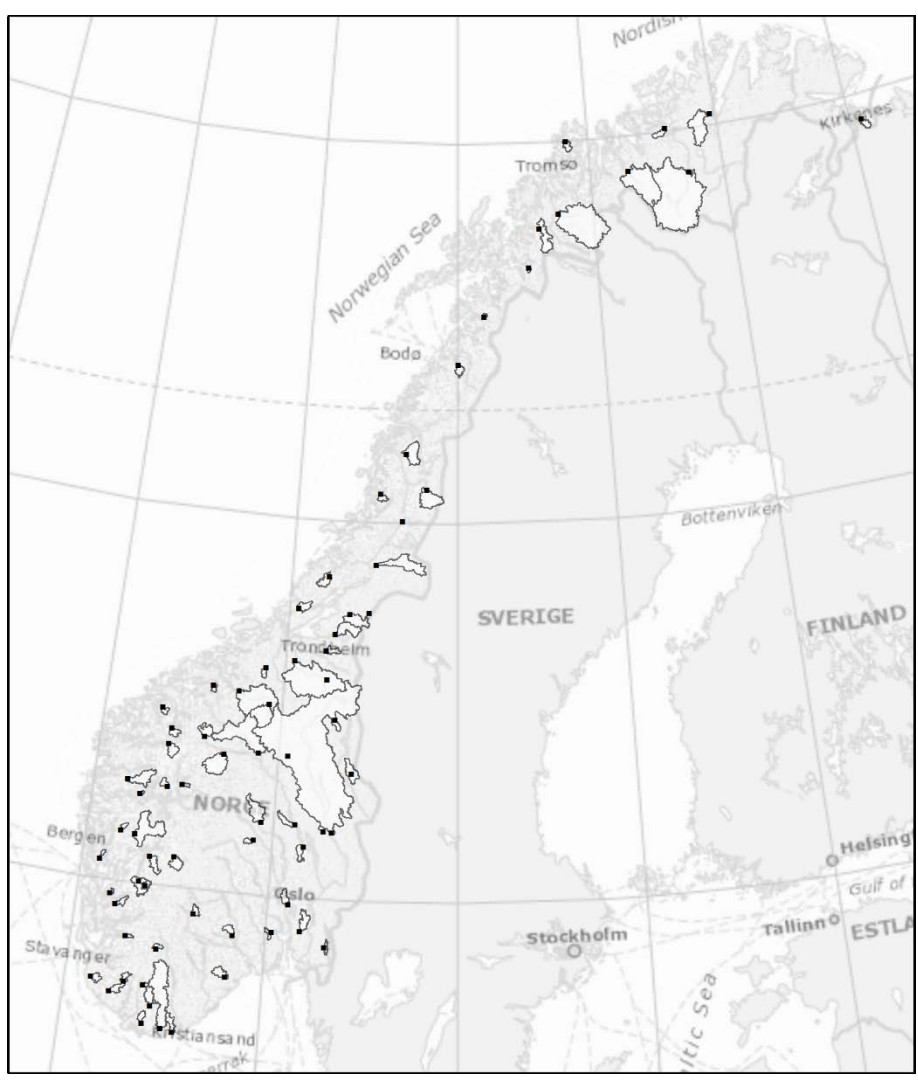

3    Fig 3



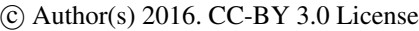

3    Fig 4





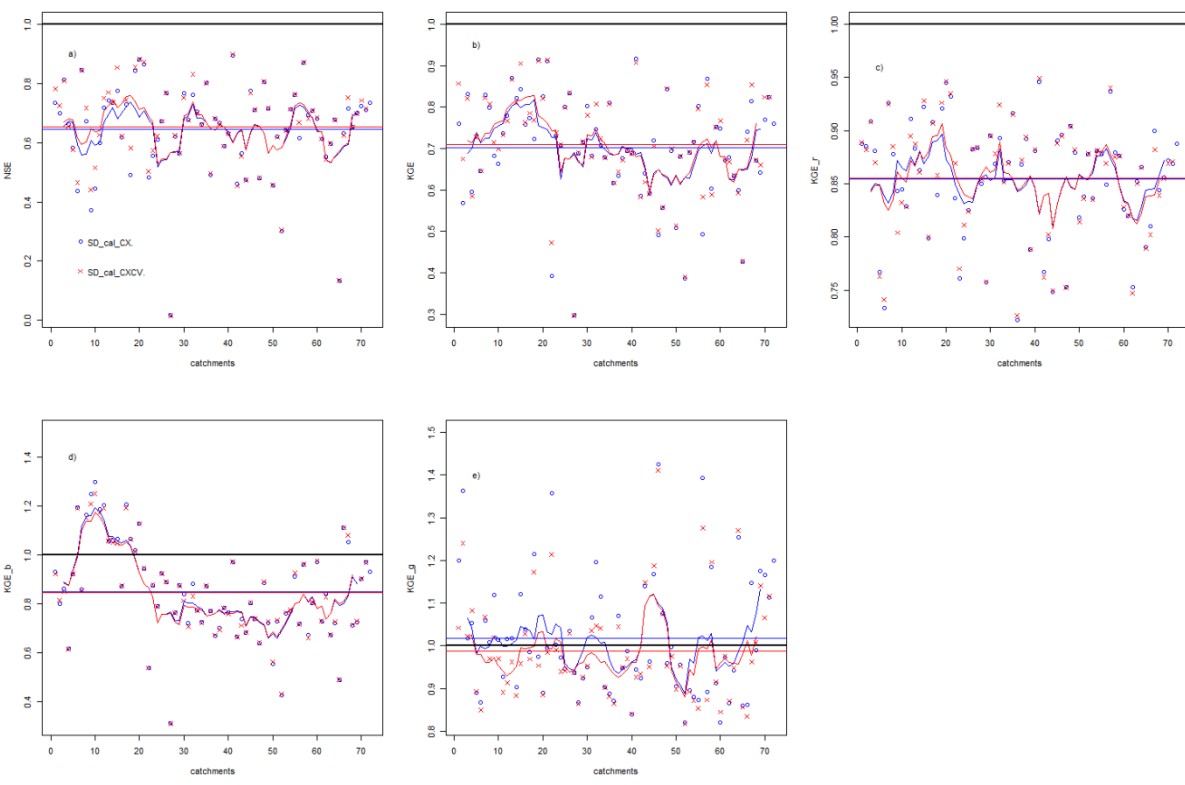

2    Fig 5





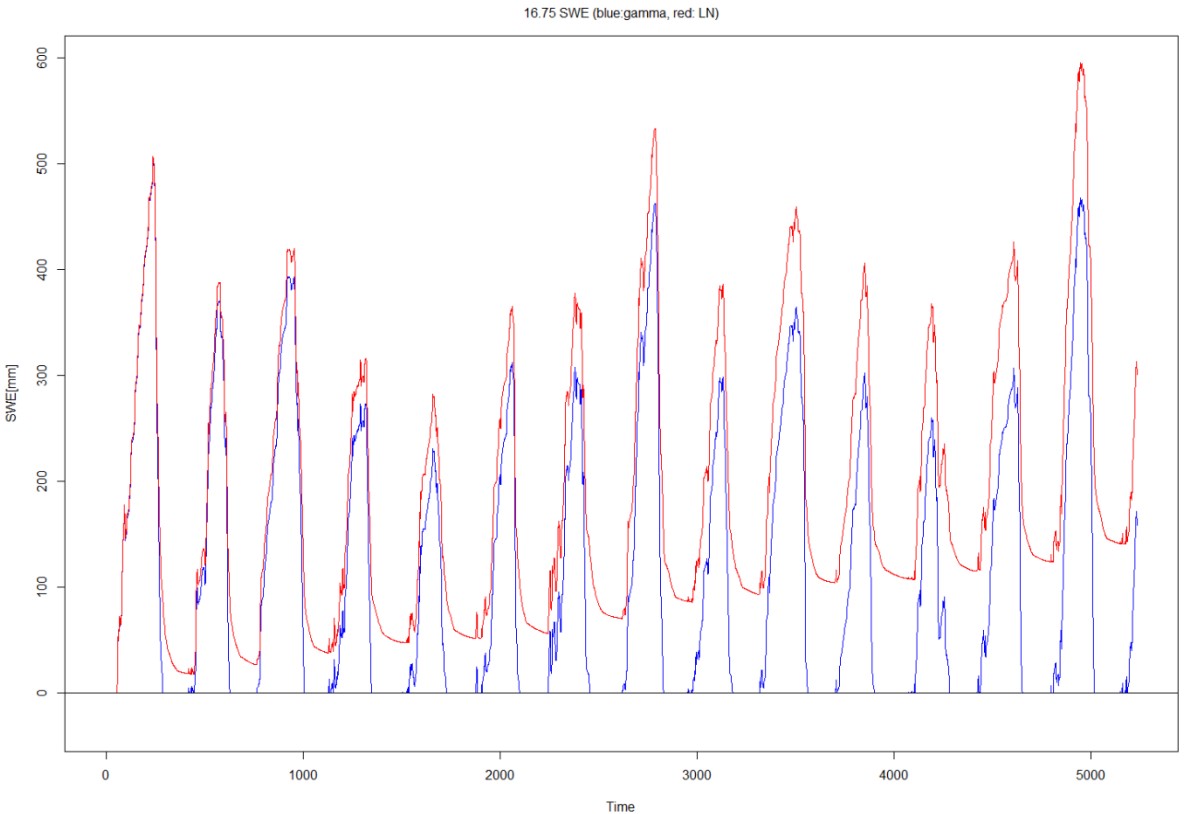

3   Fig 6



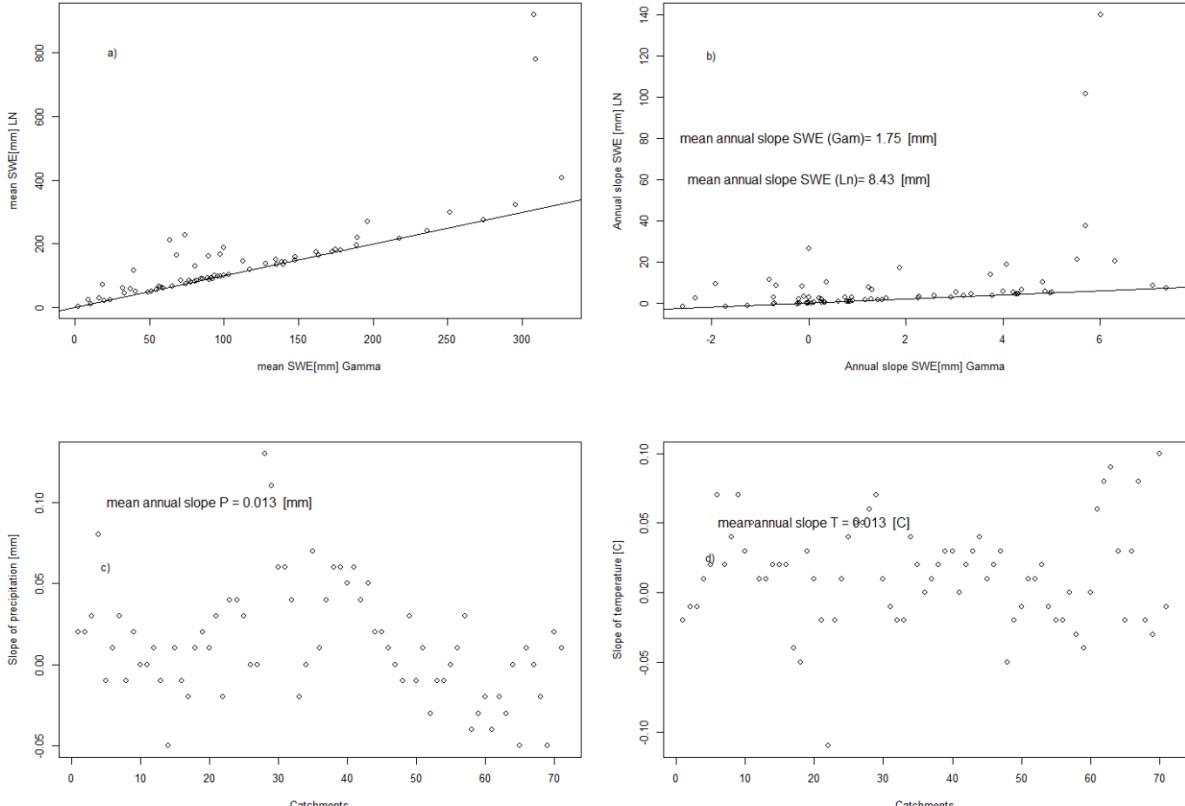

3    Fig 7





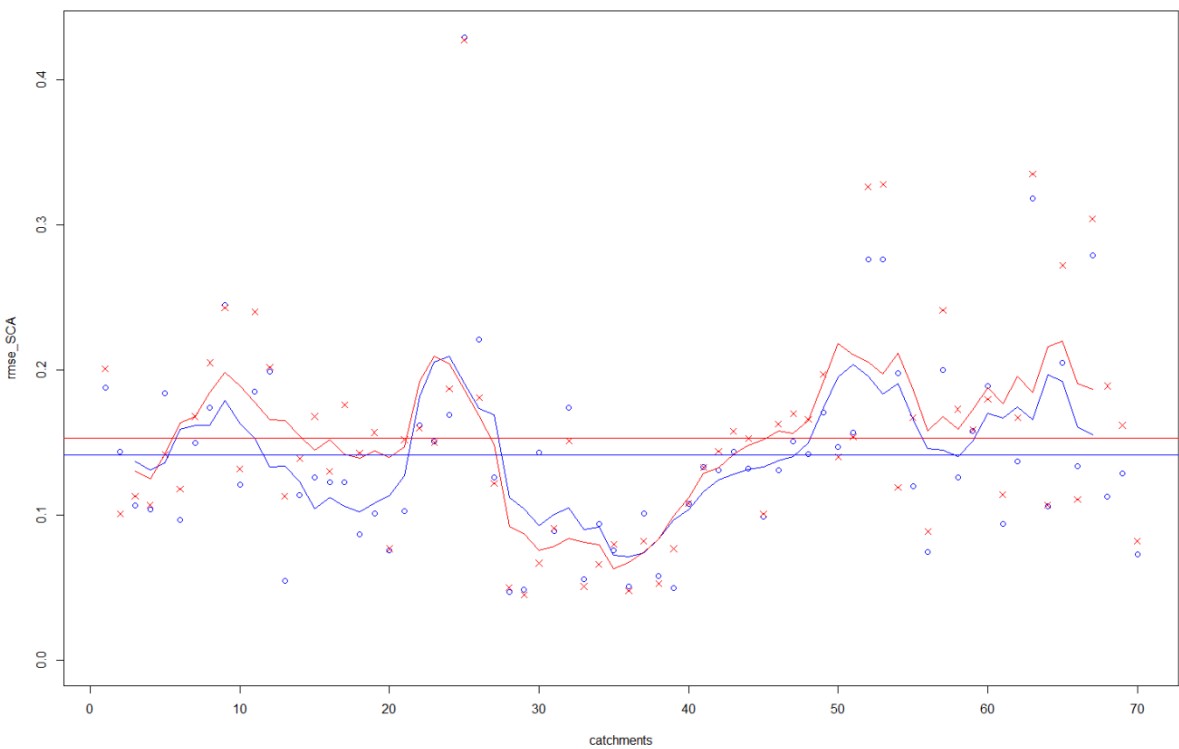

3    Fig 8





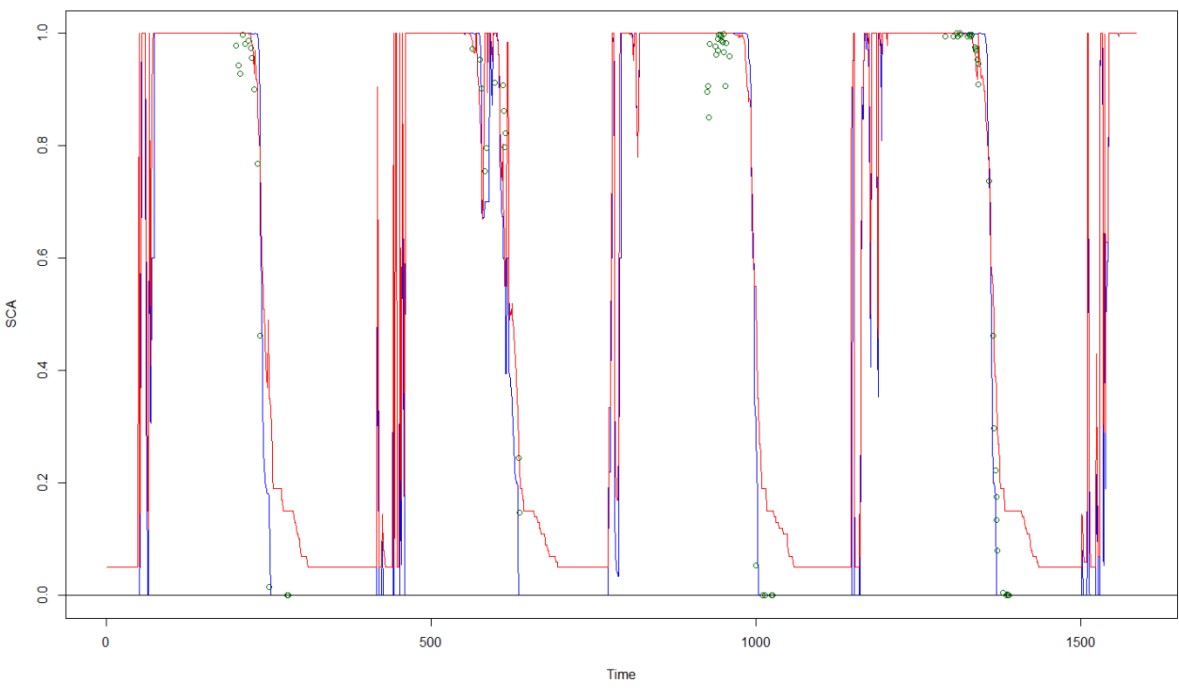

3    Fig 9





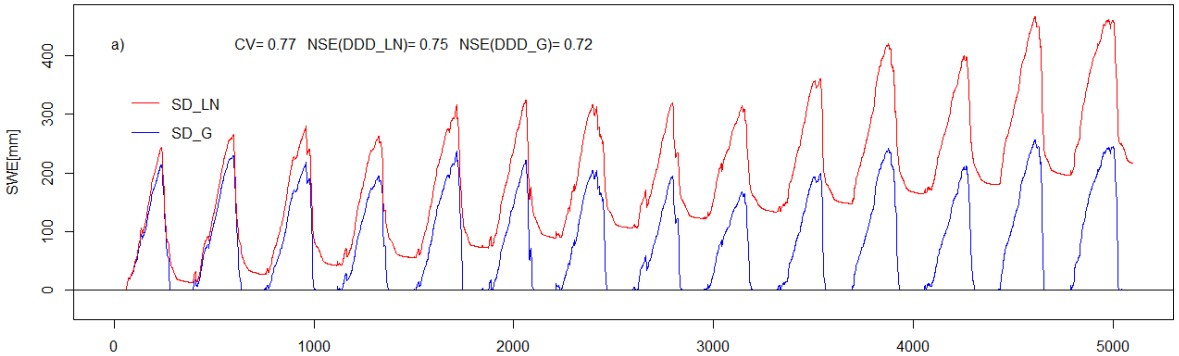

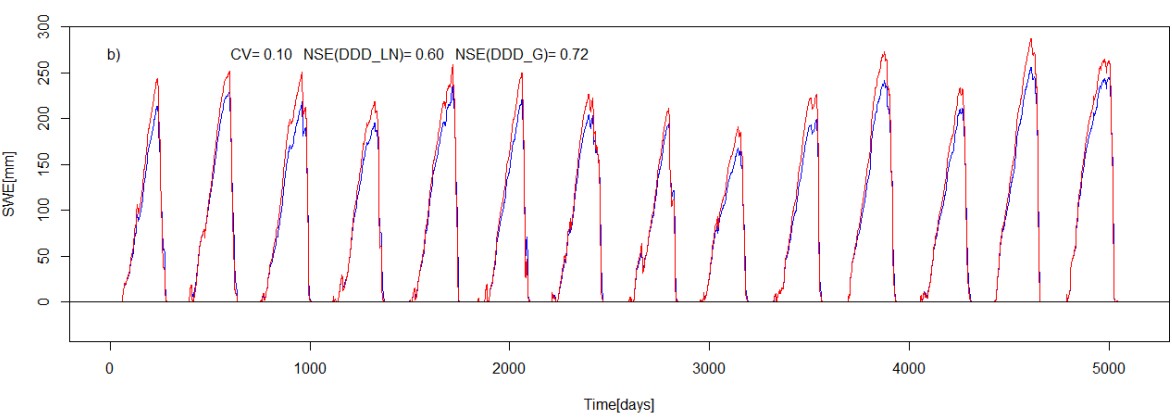

2    Fig 10