# Peer review of "A model for the spatial distribution of snow water equivalent parameterised from the spatial variability of precipitation Thomas Skaugen¹ and Ingunn H. Weltzien¹,2 \* [1] {Norwegian Water Resources and energy Directorate, P.O. Box 5091, Maj. 0301 Oslo, Norway.} [2] {Department of Geosciences, University of Oslo} Correspondence to: T. Skaugen (ths@nve.no) \*now at Norconsult AS, P.O. Box 626, 1303,"

_The Cryosphere, 2016_

## Referee Comment (RC1) · Anonymous Referee #1 · 28 Mar 2016

General comments

The manuscript evaluates an approach which combines a statistical model for the spatial distribution of SWE with a simple rainfall-runoff model (DDD model). The approach is evaluated in terms of runoff model efficiency and MODIS snow cover area in 71 basins in Norway. The results are also compared with results of existing model used for operational forecasting. The results show that the runoff model efficiency of the new model decreases only slightly, but the snow simulations are significantly improved.

The paper builds on results developed by authors in the past. The main novel contribution is the combination of existing rainfall runoff model and a model which estimates the spatial probability density function (PDF) of snow water equivalent (SWE) as a dynamic

gamma distribution. The context of the research, however, is not clearly formulated in the introduction. The basic assumptions and previous literature on the use of PDF of SWE is not clearly presented, nor the difference to SWE modelling based on simple degree-day or more sophisticated physically based snow modelling. I think this is crucial, because introduction is in many parts very technical (e.g. includes also some equations), but for readers (not familiar with such approach) it is very difficult to understand the meaning and main points of the approach and terminology. I would suggest to clearly outline the approach and also present literature which combines such statistical models with rainfall runoff modeling in the past. In the methodology some basic outline would be also useful (e.g. some schematics how the snow accumulation and melt is modelled by the approach). Moreover the results might be elaborated in more thorough way (including figures). I agree that using a large sample of basins is important, but the results do not show much of the value of such large dataset. It will be interesting, for example, to stratify the basins in the figures according mean elevation, size, or some other characteristics to show some more information than just the efficiency. It is not very clear, why the improved snow simulations do not result in better runoff simulations. Some more evaluations will be interesting here.

Specific comments

1) Abstract: The applied methodology and model concept is not clearly presented (the abbreviations SD_G, LN are not very intuitive). The period used for analyses is missing.

2) Introduction: This part does not have a clear story. It mixes different topics, but does not clearly outline the research problematic and does not clearly show what the results of previous studies are. The meaning and basics behind the PDF modelling needs to be introduced on lower technical level.

3) Modeling: It is not clear whether the results show the calibration or validation period.

4) Snow cover area results. It will be interesting also to see the model performance in terms of snow cover duration.

5) Please check references. They are not always complete and consistent.

6) Table2: Which period?

7) Fig.2: A schematic would be important to understand the method, however, here it is not clear. From the Figure and caption, the meaning of a,s, F_s, etc is not clear.

---

## Referee Comment (RC2) · Anonymous Referee #2 · 19 Apr 2016

In this study a gamma distribution (SD_G) for snow distribution was implemented in a parameter parsimonious rainfall runoff model. Snow water equivalent (SWE), snow covered area (SCA) and runoff were simulated with SD_G and compared with the usage of a log-normal distribution for 71 Norwegian catchments. Both distributions simulate runoff almost similar, although the log-normal distribution artificially creates "snow towers", which leads to a positive trend in SWE. Simulations for the SCA were validated with MODIS satellite data and show that the gamma distribution performs better than the log-normal distribution.

The novelty of this study is the implementation of the already existing gamma distribution to the rainfall runoff model and testing of the model performance for catchments of

different size.

The main criticism of the paper is the fact that it is not clearly presented, which leads to unknown answers to the reader. I classify the paper with major revision.

1. At the first time it sounds contradictory, that an improved SWE simulation does not improve the model performance in runoff. As this is one major results it needs to be clearer evaluated.

2. The main novelty of this study is the implementation of SD_G to the rainfall runoff model and testing for large catchments. I would suggest including an analysis to answer some of the following research questions: In which catchments is the model performance best? Large or small catchments? High or low elevated catchments? Catchments in the south or in the north?

3. What would happen if the simulations using SD_LN were restarted each year in autumn with no snow? This would solve the problem of the "snow towers". For me it is not clear why this is not considered? At least, it should be discussed in more detail.

The quality of the figures needs to be improved. References in the text should be ordered first chronologically and then alphabetically. Also the reference list at the end of the manuscript needs to be revised because the format is not consistent (e.g. page 31 line 7-8 vs. page 31 line 10-11 vs. page 32 line 37-38).

Introduction: The introduction is very technical e.g. page 6 line 4-18 belongs more to the methods. The introduction does not have a clear story. It is not clear how you get the information of the spatial variability of the precipitation in order to estimate the parameters for SD_G.

Methods: The methods part is very detailed with a lot of formulas. For the reader it is very difficult to follow and it is not clear for which parts in the results all these formulas are necessary. You should include the period of simulation in the methods and also your runoff measurements. Where are the data from? The description of the MODIS
satellite (page 20 line 20 – page 21 line 3) belongs also to the methods and not to the results part.

Results: This part is very short compared to the methods. The authors need to evaluate runoff, SCA and SCA with respect to different characteristics (size, elevation,. . .) of the 71 catchments.

Details: Commas are sometimes missing after an equation (e.g. equation 7), also a colon before the equation (e.g. page 13 line 11).

The correct spelling is "i.e." instead of "i.e"

Page 2 line 11: ..in the already existing parameter . . ..?

Page 6 line 6: You should define the SD_LN here and not later on page 7 line 1.

Page 8 line 3-5: Include log-normal distribution, gamma distribution. . ..

Page 8 line 9: should be "changed its shape"

Page 8 line 13: Skaugen and Randen (2013)

Page 8 line 21: include the parameter for shape and scale in the text.

Page 9 line 3: "reminder"

Page 9 line 6: $\Gamma$ is not defined.

Page 9 line 11: space is missing in equation 3.

Page 10 line 16: spatial mean

Page 10 line 18: There is no straight line in Fig 1b)

Page 12 line 15: Do "units" have the same meaning as pixels or area in this context?

Page 13 line 7: delete the comma

Page 14 line 6: bracket is not closed

Page 14 line 15: I would suggest to use f_m instead of f_s for the abbreviation of snowmelt in order to be consistent with f_a (accumulation).

Page 14 line 16: delete "the same"

Page 15 line 3: "with respect to"

Page 15 line 10: why is "spatial" written in italic?

Page 15 line 13: why "left"?

Page 16 line 21: How is the correction be applied? Can you provide more details?

Page 17 line 4: I would suggest to name the cited literature. ("is found in Skaugen...")

Page 17 line 6: From Table 1 only 5 instead of 11 model parameter are bold. The explanation of the reduction of the calibrated parameter is written in the discussion of the manuscript.

Page 17 line 9: "2.6" instead of 2.5

Page 17 line 11: delete "from"

Page 17 line 18: The following procedure was conducted:

Page 18 line 20: delete "for"

Page 19 line 11: delete ")."

Page 20 line 2: What do you mean with "most catchments"? How many catchments have these "snow towers"? Is this phenomenon only observed for high elevated catchments?

Page 20 line 18: You wrote that you found 150 estimates for SCA for each catchment. In page 21 line 4 you wrote that 69 catchments have values for SCA and 2 have no SCA observations. Also why did you write in line 7 70 catchments? Please correct these inconsistencies or explain better!
Page 21 line 5: delete "for"

Table 1: On page 16 line 18 you wrote that you use temperature and precipitation lapse rates, but why are they 0 in Table 1? Additionally, I would suggest shortening the table to the most relevant parameters, because you do not use the most of the parameters in the following. Include a space between Table and 1 (page 34 line 1) Also correct "Mean elevation of catchment"

Table 2: Where does this 1.02 value come from? You wrote in the table caption, that 1 is the ideal value.

Figure 1: "Spatial mean and standard deviation of observed precip." I would additionally suggest including the parameter values of the fitted line and rename "m" on the x-axis to "mean".

Figure 2: This figure is very hard to understand. Where comes the 0.1 on the x-axis label come from?

Figure 5: Why do you include a running average over the catchments? Are they sorted by size, mean elevation,..?

Figure 6: Is your time unit days? It would be better to choose years! What does the "16.75" in the figure caption mean?

Figure 7: I would suggest changing the y limits in the figures a and b to clearer see the differences between the log-normal and gamma distribution. Is the unit of slope of regression "mm" and "C"? I think it should be mm/time and $°C$/time ($°C$/year; mm/year)

Figure 8: include the unit of the RMSE. Does this mean that the model is around 15% wrong in estimating the SCA? Do the models underestimate or overestimate the SCA? Where are the largest errors observed?

Figure 9: It is very difficult to see anything from this figure.

---

## Author Comment (AC1) · 25 May 2016

**Response to reviewer #1 to «A model for the spatial distribution of snow water equivalent parameterised from the spatial variability of precipitation" by T. Skaugen and I.H. Weltzien.**

Let us first express our gratitude for the reviewers who spend their precious time securing the quality of our research, it is very appreciated.

We have tried to break down the general comments into separate statements and will provide a response and a suggestion of correction to each of these.

General comments:

1.The context of the research, however, is not clearly formulated in the introduction.

**Response**: We agree that, at present, the introduction could be more focused. What we want to bring across is 1) that hydrological models has too many free parameters which constitutes a problem for making predictions in ungauged basins and for a changed climate. In addition 2) we want to demonstrate that that the proposed algorithm for the spatial frequency distribution of SWE which is not calibrated against streamflow is a good alternative.
**Change**: We propose to restructure and shorten the introduction in order to focus more on the two points above. We will drop the degree-day melt model as an example of a calibrated model (p.3,l.17-p.4,l.3) since it probably just confuses the issues and instead use the calibrated log-normal spatial distribution of SWE since it is directly relevant. Furthermore, the discussion of large sample hydrology (p5, l11-p.6,l3) is, perhaps not very relevant and can be dropped or alternatively moved to the discussion. The detailed description of SD_LN (p.6,l.4-18) can be moved to a subsection of the methods.

2. The basic assumptions and previous literature on the use of PDF of SWE is not clearly presented, nor the difference to SWE modelling based on simple degree-day or more sophisticated physically based snow modelling.

**Response**: In the introduction we emphasize the importance of a realistically simulated PDF of SWE (p.4, l.19-p.5, l.3) and section 2 "Methods" (p.8,l.1-14) starts with a review of the many statistical models used for the PDF of SWE. Furthermore, the topic is revisited in the discussion (p.25, l.13-p.26, l.15).
In this study we do not consider the modelling aspects of snowmelt, only the spatial distribution (PDF) of SWE. The degree-day model is a snowmelt model, and by "more sophisticated physically based snow models" we suppose R#1 refers to point models like SNOWPACK and CROCUS, which are not used for catchment modelling and are hence not relevant for this study.
**Change**: The review on PDF models for SWE in section 2 is more suitably placed in the new, more focused introduction. It is outside the scope of the paper to also discuss snowmelt and point models.

3. I would suggest to clearly outline the approach and also present literature which combines such statistical models with rainfall runoff modeling in the past. In the methodology some basic outline would be also useful (e.g. some schematics how the snow accumulation and melt is modelled by the approach).

**Response**: Both reviewers R#1 and R#2 have comments regarding the structure of the paper, and we can understand that the paper would improve with the restructuring of especially the introduction and the methods (Section 2).
**Change**: In a restructured and more focused introduction, the approach of this study will be more clearly outlined. The methods section will have an introduction, an overview, where the different steps for estimating the spatial PDF of SWE is outlined. The procedure for snowmelt is described in section 2.5 (p.16, l18-21)

4. Moreover the results might be elaborated in more thorough way (including figures). I agree that using a large sample of basins is important, but the results do not show much of the value of such large dataset. It will be

interesting, for example, to stratify the basins in the figures according mean elevation, size, or some other characteristics to show some more information than just the efficiency.

**Response**: Again, this comment is common for both R#1 and R#2, and we think this is a good point. **Change**: We will describe the results on runoff, SWE and SCA stratifying the catchments as suggested. In a preliminary analysis we found that for 26 out of the 71 catchments (37 %) the annual increase in SWE using SD_LN was more than 5mm/year. For SD_G only 7 out of 71 catchments (10 %) exhibited such a behaviour. This feature greatly influences the duration of the snow season and the mean annual duration of the snow season was 347 days using SD_LN and 228 days using SD_G. The positive trend in SWE using SD_LN could not be associated with either catchment size, mean elevation, of the catchments, or any other catchment characteristic such as fraction of lakes, bogs, forest and bare rock. The trend was, however, significant positively correlated to the parameter $\theta_{CV}$, which is, in turn, significantly positively correlated to the KGE skill score for DDD_LN. To summarize, the positive trend in SWE using SD_LN is not associated with physiographic characteristics but is due to unrealistically high values of $\theta_{CV}$ which favours the simulation of runoff at the cost of unrealistic simulations of SWE. For the simulation of SCA, the RMSE using SD_G was not significantly correlated to any catchment characteristics. The RMSE using SD_LN was found to be significant negatively correlated only to the mean elevation of the catchments implying that SCA was better simulated for high elevation catchments using SD_LN. This is consistent with our discussion that SD_LN is better at simulating the initial development of snow-free areas. The mean absolute error (MAE) of the simulation of SCA showed that, in general, for all of Norway, there is an overestimation of SCA (more so for SD_LN (6%) than SD_G (3%)). There is a regional pattern, however, in that the underestimations for both methods (SD_G and SD_LN) are found mostly for the catchments of southern west-coast of Norway. The MAE for SCA is significant positively correlated for catchment size, and many small catchments are found in this area. In the revised version of the paper we will elaborate on the results and their link to catchment characteristics as suggested above.

5. It is not very clear, why the improved snow simulations do not result in better runoff simulations. Some more evaluations will be interesting here.

**Response**: Again, this comment is common for both R#1 and R#2, and ideally one would expect improved runoff simulations when the snow is better simulated. The failure to do so, however, is not an uncommon feature for hydrological models with many free calibration parameters. In Parajka et al. (2007) they found that when the hydrological model was calibrated against snowcover data in addition to runoff, snow simulations got better, but runoff simulations deteriorated. In our own example shown in Figure 10, SD_LN performs best with respect to runoff simulations when unrealistic snow is simulated, a clear example of a model that works well with respect to runoff, but not for the right reasons. The reason for such a behavior is probably due to inadequate model structures. When the parameter for the spatial distribution of SWE in SD_LN is allowed to be optimized against runoff without physical constraints, unreasonable values for the parameter may be the result. If, however, the snow distribution is "forced" to behave realistically, given the (inadequate) model structure, the runoff simulations deteriorate quite substantially. When SD_G is used, however, we get both reasonably good runoff and snow simulations.

**Change**: We will elaborate on this in the discussion section with arguments used above.

Specific comments:

   1) Abstract: The applied methodology and model concept is not clearly presented (the abbreviations SD_G, LN are not very intuitive). The period used for analyses is missing

**Response**: Clearly the abbreviations should be spelled out. We do find it difficult, however, to see major points where improvements on the presentation can be made. The main point is that one method is calibrated against runoff and the other method is not. There are not much room for going into details on the method.

**Change**: We will spell out the abbreviations, include the period used for analysis and try to make the outline more clear.

2) Introduction: This part does not have a clear story. It mixes different topics, but does not clearly outline the research problematic and does not clearly show what the results of previous studies are. The meaning and basics behind the PDF modelling needs to be introduced on lower technical level.

**Response**: This is a similar comment to the first general comment and we agree.
**Change**: see response and change to first general comment.

3) Modeling: It is not clear whether the results show the calibration or validation period.

**Response**: That is true. The models were calibrated on data from 19858(1.9)-2000(31.8) and validated on data from 2000(1.9) -2014(31.12)
**Change**: This information will be inserted at the appropriate place (in section 2.5)

4) Snow cover area results. It will be interesting also to see the model performance in terms of snow cover duration.

**Response**:Yes, and this comment is in line with that of R#2 for Page 20 line 2: An analysis of snow cover duration will reveal how many catchments that suffers from "snow-towers" using SD_LN
**Change**: We will analyse the snowcover duration using SD_G and SD_LN, and answer both this comment and that of R#2.

5) Please check references. They are not always complete and consistent.

**Response**: Yes
**Change**: We did not find any references that where in the text and not in the reference list and vice versa, but we will edit the references in the text (correct ordering) and check the format in the reference list.

6) Table2: Which period?

**Response**: Sorry, an omission.
**Change**: Correct period (2000-2014) will be inserted

7) Fig.2: A schematic would be important to understand the method, however, here it is not clear. From the Figure and caption, the meaning of a,s, F_s, etc is not clear.

**Response**: We understand that this might be hard to grasp.
**Change**: We will elaborate further on the explanation in the text and on the figure. We suggest the following:

In the model for estimating the reduction of SCA after a melting event, we assume that areas with smallest SWE are the first to become snow free since the energy requirements for melting a column of snow is proportional to the height of snow (Dingman, 2002). Figure 2 a) shows the PDFs of melt ($f_m$, red) and accumulation ($f_a$, blue). In Figure 2 b) we have plotted the integral of the PDFs for successive intervals of SWE, so each horizontal bar represents a fractional area (see the x-axis) covered by SWE or melt values. The horizontal bars for each integrated PDF sum up to unity, i.e. the entire area covered by snow. The figures illustrates that melt values less than $X$ cover a large area (the integral of $f_m$ up to $X$, called $m$, $\int_0^X f_m = m$ in the Figure 2a) and much larger than the area of SWE values less than $X$ (the integral of $f_a$ up to $X$, called $a$, $\int_0^X f_a = a$ in Figure 2a). Consequently, the fractional area of SWE values less than $X$, $a$, becomes snow free after the melting event. In addition, there are melt values higher than $X$ that reduce the coverage of corresponding SWE values. The sum of these bars adds up to $1 - m$, and

equals the integral $\int_X^\infty f_m = 1 - m$. In total, the snow free area after a melting event is $a + 1 - m$ and is seen in Figure 2b) as the sum of the cross-hatched bars.

[Figure]

New Figure 2.

---

## Author Comment (AC2) · 25 May 2016

**Response to reviewer #2 to «A model for the spatial distribution of snow water equivalent parameterised from the spatial variability of precipitation" by T. Skaugen and I.H. Weltzien.**

Let us first express our gratitude for the reviewers who spend their precious time securing the quality of our research, it is very appreciated.

General comments:
1. At the first time it sounds contradictory, that an improved SWE simulation does not improve the model performance in runoff. As this is one major results it needs to be clearer evaluated.

**Response**: R#1 had a similar comment (general comment #5), please see the response and suggested change.

2. The main novelty of this study is the implementation of SD_G to the rainfall runoff model and testing for large catchments. I would suggest including an analysis to answer some of the following research questions: In which catchments is the model performance best? Large or small catchments? High or low elevated catchments? Catchments in the south or in the north?

**Response:** R#1 had a similar comment (general comment #4), please see the response and suggested change.

3. What would happen if the simulations using SD_LN were restarted each year in autumn with no snow? This would solve the problem of the "snow towers". For me it is not clear why this is not considered? At least, it should be discussed in more detail.

**Response**: Such a procedure would solve the immediate problem of the snow towers, but we would still be left with a routine for the spatial snow distribution that did not work properly and/or is conceptually wrong. The coming and going of snow in a catchment is a process governed by the climate. Sometimes, in Norwegian catchments, snow survives the summer and other times it does not. Our ambition must be to have models that simulates this behavior without relying on manually updating the snow reservoir (which is not a trivial task since the other reservoirs/states in the hydrological model have to be updated as well). **Change**: No suggested change, we have already discussed this in some detail at p.22,l.12-p.21,l.11)

4. The quality of the figures needs to be improved. References in the text should be ordered first chronologically and then alphabetically. Also the reference list at the end of the manuscript needs to be revised because the format is not consistent (e.g. page 31 line 7-8 vs. page 31 line 10-11 vs. page 32 line 37-38).

**Response**: Noted
**Change**: We will improve figures as suggested below and edit the references in the text and check the format in the reference list.

Introduction: The introduction is very technical e.g. page 6 line 4-18 belongs more to the methods. The introduction does not have a clear story. It is not clear how you get the information of the spatial variability of the precipitation in order to estimate the parameters for SD_G.

**Response**: R#1 had a similar comment (general comment #1), please see the response and suggested change.
**Change**: We will make sure that the information on how the spatial variability of precipitation is obtained is clearly explained.

Methods: The methods part is very detailed with a lot of formulas. For the reader it is very difficult to follow and it is not clear for which parts in the results all these formulas are necessary. You should include the period of simulation in the methods and also your runoff measurements. Where are the data from? The description of the MODIS satellite (page 20 line 20 – page 21 line 3) belongs also to the methods and not to the results part.

**Response**: R#1 had a similar comment (general comment #3), please see the response and suggested change. The results are obtained by, at all times, having estimates of the spatial moments (the spatial mean and variance of SWE) in order to estimate the spatial PDF, so all the formulas are necessary. The precipitation data are from the Norwegian meteorological institute, whereas the runoff data are from Norwegian water resources and Energy Directorate (NVE).
**Change**: We will start this section with have an introduction, an overview, where the different steps for estimating the spatial PDF of SWE is outlined. This will help the reader to get an overall understanding of the method without detailed study of the equations. We will include the description of the MODIS data in the methods.

Results: This part is very short compared to the methods. The authors need to evaluate runoff, SCA and SCA with respect to different characteristics (size, elevation,. . .) of the 71 catchments.

**Response**: R#1 had a similar comment (general comment #4), please see the response and suggested change.

Specific comments:

Commas are sometimes missing after an equation (e.g. equation 7), also a colon before the equation (e.g. page 13 line 11).

**Response** :Noted
**Change**: It will be changed

The correct spelling is "i.e." instead of "i.e"

**Response** :Noted
**Change**: It will be changed

Page 2 line 11: ..in the already existing parameter . . ..?

**Response**: Noted
**Change**: perhaps just delete "already"

Page 6 line 6: You should define the SD_LN here and not later on page 7 line 1.

**Response** :Noted
**Change**: It will be changed

Page 8 line 3-5: Include log-normal distribution, gamma distribution. . ..

**Response** :Noted
**Change**: It will be changed

Page 8 line 9: should be "changed its shape"

**Response** :Noted
**Change**: It will be changed

Page 8 line 13: Skaugen and Randen (2013)

**Response** :Noted
**Change**: It will be changed

Page 8 line 21: include the parameter for shape and scale in the text.

**Response** :Noted
**Change**: It will be changed

Page 9 line 3: "reminder"

**Response** :Noted
**Change**: It will be changed

Page 9 line 6: Γ is not defined.

**Response**: Noted
**Change**: The gamma function will be defined

Page 9 line 11: space is missing in equation 3.

**Response** :Noted
**Change**: It will be changed

Page 10 line 16: spatial mean

**Response** :Noted
**Change**: It will be changed

Page 10 line 18: There is no straight line in Fig 1b)

**Response**: Agreed
**Change**: We will replace "does" with "will".

Page 12 line 15: Do "units" have the same meaning as pixels or area in this context?

**Response**: No, a unit is an amount of SWE (it is later defined as 0.1 mm)
**Change**: We will include the notation [mm], when the units are first mentioned (p.9,l.4)

Page 13 line 7: delete the comma

**Response** :Noted
**Change**: It will be changed

Page 14 line 6: bracket is not closed

**Response** :Noted
**Change**: It will be changed

Page 14 line 15: I would suggest to use f_m instead of f_s for the abbreviation of snowmelt in order to be consistent with f_a (accumulation).

**Response**: A good idea
**Change**: It will be changed, also in the Figure 2.

Page 14 line 16: delete "the same"

**Response** :Noted
**Change**: It will be changed

Page 15 line 3: "with respect to"

**Response** :Noted
**Change**: It will be changed

Page 15 line 10: why is "spatial" written in italic?

**Response**: Just to emphasize that it is spatial frequency distributions such that the frequencies and their integral can be seen as areas.
**Change**: It can be removed

Page 15 line 13: why "left"?

**Response**: They will become snowfree
**Change**: We will reformulate

Page 16 line 21: How is the correction be applied? Can you provide more details?

**Response**: Precipitation is increased or decreased by multiplying the amount with a constant.
**Change**: we will reformulate the sentence

Page 17 line 4: I would suggest to name the cited literature. ("is found in Skaugen. . .")

**Response** :Noted
**Change**: It will be changed

Page 17 line 6: From Table 1 only 5 instead of 11 model parameter are bold. The explanation of the reduction of the calibrated parameter is written in the discussion of the manuscript.

**Response**: 11 parameters can potentially be calibrated. In this study only 5 parameters are calibrated either using V1 or V2 (parameters in bold in Table 1).
**Change**: The explanation of the reduction of the calibrated parameters is written in the methods section (Sub sect. 2.5 (should be 2.6)). We will change the table caption of Table 1 to emphasize that in this study only 5 parameters are calibrated.

Page 17 line 9: "2.6" instead of 2.5

**Response** :Noted
**Change**: It will be changed

Page 17 line 11: delete "from"

**Response** :Noted
**Change**: It will be changed

Page 17 line 18: The following procedure was conducted:

**Response** :Noted
**Change**: It will be changed

Page 18 line 20: delete "for"

**Response** :Noted
**Change**: It will be changed

Page 19 line 11: delete ")."

**Response** :Noted
**Change**: It will be changed

Page 20 line 2: What do you mean with "most catchments"? How many catchments have these "snow towers"? Is this phenomenon only observed for high elevated catchments?

**Response**: We agree that the term "most catchments" is not very precise. The high mean annual slope of SWE using SD_LN was the cause of such a statement.

**Change**: In the stratified analysis of the catchments with respect to efficiency for runoff simulation SWE and SCA we will include a quantification of such behavior and see if it is related to mean elevation, catchment size etc. (see response and change to R#1, general comment #4)

Page 20 line 18: You wrote that you found 150 estimates for SCA for each catchment. In page 21 line 4 you wrote that 69 catchments have values for SCA and 2 have no SCA observations. Also why did you write in line 7 70 catchments? Please correct these inconsistencies or explain better!

**Response**: Sorry, a typo. There are 71 catchments. Only 69 catchments have estimated SCA
**Change**: We will change the numbers in P.20,l.17 and on P.21,L.7.

Page 21 line 5: delete "for"

**Response** :Noted
**Change**: It will be changed

Table 1: On page 16 line 18 you wrote that you use temperature and precipitation lapse rates, but why are they 0 in Table 1? Additionally, I would suggest shortening the table to the most relevant parameters, because you do not use the most of the parameters in the following. Include a space between Table and 1 (page 34 line 1) Also correct "Mean elevation of catchment"

**Response**: They are set to zero since they are not used. Unless the editor wishes otherwise, we would like to keep the table as it is since it is complete for the DDD model. Just having a subset of the table would demand an additional paragraph explaining the other parameters.
**Change**: Space will be inserted and also the correct spelling of "catchment".

Table 2: Where does this 1.02 value come from? You wrote in the table caption, that 1 is the ideal value.

**Response**: 1 is indeed the ideal value but the variability error is allowed to be more than 1 (signifies higher variability than the observed series), see Kling et al. (2012), full reference is found in the paper.
**Change**: No change.

Figure 1: "Spatial mean and standard deviation of observed precip." I would additionally suggest including the parameter values of the fitted line and rename "m" on the x-axis to "mean".

**Response** :Noted
**Change**: It will be changed

Figure 2: This figure is very hard to understand. Where comes the 0.1 on the x-axis label come from?

**Response**:R#1 had the same comment (specific commet #7). Since we deal with spatial frequency distributions, one must think of the frequencies as number of locations with a given SWE value. The x-axis shows the number of units, so we have to multiply with the unit value (0.1 mm) in order to have mm.
**Change**: we will make a new Figure 2 and elaborate on the explanation (see response to R#1, specific comment #7)

Figure 5: Why do you include a running average over the catchments? Are they sorted by size, mean elevation,..?

**Response**: The running mean was included to improve readability. They are not sorted by size, elevation but geographically, starting with central southern Norway, moving along the coast to the north.
**Change**: A new analysis of the results will be conducted and the figure will be replaced (see response and change to R#1, general comment #4).

Figure 6: Is your time unit days? It would be better to choose years! What does the "16.75" in the figure caption mean?

**Response**: Yes. "16.75 " is the identification of the catchment"

**Change**: We can add proper axis labels and remove "16.75"

Figure 7: I would suggest changing the y limits in the figures a and b to clearer see the differences between the log-normal and gamma distribution. Is the unit of slope of regression "mm" and "C"? I think it should be mm/time and _C/time (_C/year; mm/year)

**Response**: Agreed, to both comments
**Change**: We will change the y limits, and have proper units (mm/year and °C/year)

Figure 8: include the unit of the RMSE. Does this mean that the model is around 15% wrong in estimating the SCA? Do the models underestimate or overestimate the SCA? Where are the largest errors observed?

**Response**: We can include the unit and yes, the models are around 15% wrong in estimating SCA.
**Change**: In a more stratified analysis of the results we will answer the questions posed by the reviewer.(see response and change to R#1, general comment #4)

Figure 9: It is very difficult to see anything from this figure.

**Response**: The figure should have proper labels, but we do not see why it is so difficult to read the figure. Red and blue are simulated values of SCA and the green circles represents observed SCA, just as the figure captions says.
**Change**: We will add proper labels with units to the figure

---

## Editor Comment (EC1) · R. D. Brown (Editor) · 26 May 2016

In response to the authors' query to the Editor, I agree with reviewer 2's point about Table 1 and do not think it very helpful to set a value to zero to indicate it is not used. Overall I find this Table rather confusing with "standard value" entered in the "method of estmation column" followed by either a blank or a number in the "value" column. My suggestion is to only include columns 1 and 2, with the few cases of additional information from the "value" column included in the "comment" column (which I would rename "Description"). You can also add information like "Derived from GIS" in the description of the model parameters.

RB (Ed)

---

## Author Response (AR1)

**Response to reviewer #1 to «A model for the spatial distribution of snow water equivalent parameterised from the spatial variability of precipitation" by T. Skaugen and I.H. Weltzien.**

Let us first express our gratitude for the reviewers who spend their precious time securing the quality of our research, it is very appreciated.

We have tried to break down the general comments into separate statements and will provide a response and a suggestion of correction to each of these.

In the marked-up MS new text is marked in red and moved text is marked with green. Unfortunately, the word version is Norwegian, so "Slettet" means deleted and "Flyttet" means moved. We hope this is not too inconvenient.

General comments:

1.The context of the research, however, is not clearly formulated in the introduction.

**Response**: We agree that, at present, the introduction could be more focused. What we want to bring across is 1) that hydrological models has too many free parameters which constitutes a problem for making predictions in ungauged basins and for a changed climate. In addition 2) we want to demonstrate that that the proposed algorithm for the spatial frequency distribution of SWE which is not calibrated against streamflow is a good alternative.
**Change**: We have restructured and shortened the introduction in order to focus more on the two points above. We have dropped the degree-day melt model as an example of a calibrated model since it probably just confuses the issues. Furthermore, the discussion of large sample hydrology is dropped. The detailed description of SD_LN is moved to subsection 2.3 (p.17.l1-19 in marked MS) of the methods section. We have also included the review of the spatial PDF of SWE used in hydrological modelling (p5.l.11-20, p.6.,l.1-4, in marked MS), originally placed in the methods section, in the introduction.

2. The basic assumptions and previous literature on the use of PDF of SWE is not clearly presented, nor the difference to SWE modelling based on simple degree-day or more sophisticated physically based snow modelling.

**Response**: In the introduction we emphasize the importance of a realistically simulated PDF of SWE (p.4, l.19-p.5, l.3) and section 2 "Methods" (p.8,l.1-14) starts with a review of the many statistical models used for the PDF of SWE. Furthermore, the topic is revisited in the discussion (p.25, l.13-p.26, l.15).
In this study we do not consider the modelling aspects of snowmelt, only the spatial distribution (PDF) of SWE. The degree-day model is a snowmelt model, and by "more sophisticated physically based snow models" we suppose R#1 refers to point models like SNOWPACK and CROCUS, which are not used for catchment modelling and are hence not relevant for this study.

**Change**: The review on PDF models for SWE in section 2 is more suitably placed in the new, more focused introduction (p5.l.11-20, p.6.,l.1-4, in marked MS). It is outside the scope of the paper to also discuss snowmelt and point models.

3. I would suggest to clearly outline the approach and also present literature which combines such statistical models with rainfall runoff modeling in the past. In the methodology some basic outline would be also useful (e.g. some schematics how the snow accumulation and melt is modelled by the approach).

**Response**: Both reviewers R#1 and R#2 have comments regarding the structure of the paper, and we can understand that the paper would improve with the restructuring of especially the introduction and the methods (Section 2).

**Change**: In the restructured and more focused introduction, the approach of this study is more clearly outlined. The methods section hasl have an introduction, an overview (p.7,l.20-22, p.8 and p.9,l.1-3 in marked MS), where the different steps for estimating the spatial PDF of SWE is outlined. The procedure for snowmelt is described in section 2.3 (p.16, l19-21 in marked MS)

4. Moreover the results might be elaborated in more thorough way (including figures). I agree that using a large sample of basins is important, but the results do not show much of the value of such large dataset. It will be interesting, for example, to stratify the basins in the figures according mean elevation, size, or some other characteristics to show some more information than just the efficiency.

**Response**: Again, this comment is common for both R#1 and R#2, and we think this is a good point.
**Change**: We describe the results on runoff, SWE and SCA stratifying the catchments as suggested. We hav included a new table (Table 3, p43 in marked MS) showing significant correlations between the results and catchment characteristics (CCs). When the results for Runoff, SWE, SCA and snow cover duration are presented, we also present significant correlations between results and CCs. (p.20, l18-19 ,p.21, p22,p23,p24,l1-9, in marked MS) A new Figure (Fig.9) is included that shows the mean snowcover duration using the two models. In figures 5, 8, and 9, the catchments are now organised geographically.

5. It is not very clear, why the improved snow simulations do not result in better runoff simulations. Some more evaluations will be interesting here.

**Response**: Again, this comment is common for both R#1 and R#2, and ideally one would expect improved runoff simulations when the snow is better simulated. The failure to do so, however, is not an uncommon feature for hydrological models with many free calibration parameters. In Parajka et al. (2007) they found that when the hydrological model was calibrated against snowcover data in addition to runoff, snow simulations got better, but runoff simulations deteriorated. In our own example shown in Figure 10, SD_LN performs best with respect to runoff simulations when unrealistic snow is simulated, a clear example of a model that works well with respect to runoff, but not for the right reasons. The reason for such a behavior is probably due to inadequate model structures. When the parameter for the spatial distribution of SWE in SD_LN is allowed to be optimized against runoff without physical constraints, unreasonable values for the parameter may be the result. If, however, the snow distribution is "forced" to behave realistically, given the (inadequate) model structure, the runoff simulations deteriorate quite substantially. When SD_G is used, however, we get both reasonably good runoff and snow simulations.

**Change**: We have elaborated on this in the discussion section with arguments used above (p.25, l.7-15 in marked MS).

Specific comments:

  1) Abstract: The applied methodology and model concept is not clearly presented (the abbreviations SD_G, LN are not very intuitive). The period used for analyses is missing

**Response**: Clearly the abbreviations should be spelled out. We do find it difficult, however, to see major points where improvements on the presentation can be made. The main point is that one method is calibrated against runoff and the other method is not. There are not much room for going into details on the method.
**Change**: We have spelled out the abbreviations and included the period used for analysis and tried to make the outline more clear (p.2 in marked MS).

  2) Introduction: This part does not have a clear story. It mixes different topics, but does not clearly outline the research problematic and does not clearly show what the results of previous studies are. The meaning and basics behind the PDF modelling needs to be introduced on lower technical level.

**Response**: This is a similar comment to the first general comment and we agree.
**Change**: see response and change to first general comment.

  3) Modeling: It is not clear whether the results show the calibration or validation period.

**Response**: That is true. The models were calibrated on data from 19858(1.9)-2000(31.8) and validated on data from 2000(1.9) -2014(31.12)
**Change**: This information is included in section 2.4 (p.18, l.19-29 in marked MS)

  4) Snow cover area results. It will be interesting also to see the model performance in terms of snow cover duration.

**Response**:Yes, and this comment is in line with that of R#2 for Page 20 line 2: An analysis of snow cover duration will reveal how many catchments that suffers from "snow-towers" using SD_LN
**Change**: We have analysed the snowcover duration using SD_G and SD_LN, see Figure 9 and in sect 3. , (p.24, l.3-9 in marked MS) and in sect. 4 (p.27, l.4-12 in marked MS)

  5) Please check references. They are not always complete and consistent.

**Response**: Yes

**Change**: We have edited the references in the text (consistent ordering) and in the reference list (correct format).

6) Table2: Which period?

**Response**: Sorry, an omission.
**Change**: We have inserted the correct period (2000-2014) p.41 in marked MS.

7) Fig.2: A schematic would be important to understand the method, however, here it is not clear. From the Figure and caption, the meaning of a,s, F_s, etc is not clear.

**Response**: We understand that this might be hard to grasp.
**Change**: We have elaborated further on the explanation in the text and on the figure. See new Figure 2 at p. 47 in marked MS and p.15, l. 1-15 in marked MS.

**Response to reviewer #2 to «A model for the spatial distribution of snow water equivalent parameterised from the spatial variability of precipitation" by T. Skaugen and I.H. Weltzien.**

General comments:
1. At the first time it sounds contradictory, that an improved SWE simulation does not improve the model performance in runoff. As this is one major results it needs to be clearer evaluated.

**Response**: R#1 had a similar comment (general comment #5), please see the response and change.

2. The main novelty of this study is the implementation of SD_G to the rainfall runoff model and testing for large catchments. I would suggest including an analysis to answer some of the following research questions: In which catchments is the model performance best? Large or small catchments? High or low elevated catchments? Catchments in the south or in the north?

**Response:** R#1 had a similar comment (general comment #4), please see the response and change.

3. What would happen if the simulations using SD_LN were restarted each year in autumn with no snow? This would solve the problem of the "snow towers". For me it is not clear why this is not considered? At least, it should be discussed in more detail.

**Response**: Such a procedure would solve the immediate problem of the snow towers, but we would still be left with a routine for the spatial snow distribution that did not work properly and/or is conceptually wrong. The coming and going of snow in a catchment is a process governed by the climate. Sometimes, in Norwegian catchments, snow survives the summer and other times it does not. Our ambition must be to have models that simulates this behavior without relying on manually updating the snow reservoir (which is not a trivial task since the other reservoirs/states in the hydrological model have to be updated as well).
**Change**: No change, we have already discussed this in some detail at p.25,l.15-p.26,1.1-16 in marked MS.

4. The quality of the figures needs to be improved. References in the text should be ordered first chronologically and then alphabetically. Also the reference list at the end of the manuscript needs to be revised because the format is not consistent (e.g. page 31 line 7-8 vs. page 31 line 10-11 vs. page 32 line 37-38).

**Response**: Noted
**Change**: We have improved the Figures. Figures 1,2, 5-11 are all new and we have edited the references in the text and in the reference list.

Introduction: The introduction is very technical e.g. page 6 line 4-18 belongs more to the methods. The introduction does not have a clear story. It is not clear how you get the information of the spatial variability of the precipitation in order to estimate the parameters for SD_G.

**Response**: R#1 had a similar comment (general comment #1), please see the response and change.

**Change**: Information on how the spatial variability of precipitation is obtained is explained in sect 2.4,
p.18, l.11-15 in the marked MS.
Methods: The methods part is very detailed with a lot of formulas. For the reader it is very difficult to follow and it is not
clear for which parts in the results all these formulas are necessary. You should include the period of simulation in the
methods and also your runoff measurements. Where are the data from? The description of the MODIS satellite (page
20 line 20 – page 21 line 3) belongs also to the methods and not to the results part.
**Response**: R#1 had a similar comment (general comment #3), please see the response and change. The
results are obtained by, at all times, having estimates of the spatial moments (the spatial mean and
variance of SWE) in order to estimate the spatial PDF, so all the formulas are necessary. The
precipitation data are from the Norwegian meteorological institute, whereas the runoff data are from
Norwegian water resources and Energy Directorate (NVE).
**Change**: In formation on the data and periods (including the MODIS images) are found in sect 2.4 in the
new MS.
Results: This part is very short compared to the methods. The authors need to evaluate runoff, SCA and SCA with
respect to different characteristics (size, elevation,. . .) of the 71 catchments.
**Response**: R#1 had a similar comment (general comment #4), please see the response and change.
Specific comments:
Commas are sometimes missing after an equation (e.g. equation 7), also a colon before the equation (e.g. page 13 line
11).
**Response** :Noted
**Change**: It is changed, see various places in marked MS
The correct spelling is "i.e." instead of "i.e"
**Response** :Noted
**Change**: It is changed, see various places in marked MS
Page 2 line 11: ..in the already existing parameter . . ..?
**Response**: Noted
**Change**: "already" is deleted : see p.2, l. 12 in marked MS
Page 6 line 6: You should define the SD_LN here and not later on page 7 line 1.
**Response** :Noted
**Change**: This section has been restructured. SD_LN is defined in the abstract and  on p.7,l.1 in the
marked MS

Page 8 line 3-5: Include log-normal distribution, gamma distribution. . ..

**Response** :Noted
**Change**: It is changed , see p.5. l.14-19 in marked MS

Page 8 line 9: should be "changed its shape"

**Response** :Noted
**Change**: It is changed, see p.6, l.1 in marked MS

Page 8 line 13: Skaugen and Randen (2013)

**Response** :Noted
**Change**: It is changed, p.7, l.21 in marked MS

Page 8 line 21: include the parameter for shape and scale in the text.

**Response** :Noted
**Change**: It is changed, see p.8, l.1 in marked MS

Page 9 line 3: "reminder"

**Response** :Noted
**Change**: It is actually correct with "remainder", no change.

Page 9 line 6: Γ is not defined.

**Response**: Noted
**Change**: The gamma function is defined , see p.9, l.11. in marked MS

Page 9 line 11: space is missing in equation 3.

**Response** :Noted
**Change**: It is changed, p.9, l.15 in marked MS

Page 10 line 16: spatial mean

**Response** :Noted
**Change**: It is changed, see p.11, l.2. in marked MS

Page 10 line 18: There is no straight line in Fig 1b)

**Response**: Agreed
**Change**: We have replaced "does" with "will". See p.11, l.5 in marked MS

Page 12 line 15: Do "units" have the same meaning as pixels or area in this context?

**Response**: No, a unit is an amount of SWE (it is later defined as 0.1 mm)
**Change**: We have included the notation [mm], when the units are first mentioned (p.8, l.11 in marked MS)

Page 13 line 7: delete the comma

**Response** :Noted
**Change**: It is changed, see p.13, l.5 in marked MS

Page 14 line 6: bracket is not closed

**Response** :Noted
**Change**: It is changed, se p. 14, l.3 in marked MS

Page 14 line 15: I would suggest to use f_m instead of f_s for the abbreviation of snowmelt in order to be consistent with f_a (accumulation).

**Response**: A good idea
**Change**: It is changed, see p.14,l.12, p.15, p.16, l. 2 in marked MS. And  new Figure 2., p. 47 in marked MS

Page 14 line 16: delete "the same"

**Response** :Noted
**Change**: It is changed, see p.14, l.13 in marked MS.

Page 15 line 3: "with respect to"

**Response** :Noted
**Change**: It is rewritten, see p.15, l.18 in marked MS

Page 15 line 10: why is "spatial" written in italic?

**Response**: Just to emphasize that it is spatial frequency distributions such that the frequencies and their integral can be seen as areas.
**Change**: This part has been rewritten, see p.15 in marked MS

Page 15 line 13: why "left"?

**Response**: They will become snowfree
**Change**: This part has been rewritten, see p.15 in marked MS

Page 16 line 21: How is the correction be applied? Can you provide more details?

**Response**: Precipitation is increased or decreased by multiplying the amount with a constant.

**Change**: This part has been rewritten, see p.16, l.18-19 in marked MS.

Page 17 line 4: I would suggest to name the cited literature. ("is found in Skaugen. . .")

**Response** :Noted
**Change**: It is changed, see p.18, l.1 in marked MS

Page 17 line 6: From Table 1 only 5 instead of 11 model parameter are bold. The explanation of the reduction of the calibrated parameter is written in the discussion of the manuscript.

**Response**: 11 parameters can potentially be calibrated. In this study only 5 parameters are calibrated either using V1 or V2 (parameters in bold in Table 1).
**Change**: It is changed, see p.18, l.2-5 in marked MS and in the cation for Table 1, p. 38 in marked MS

Page 17 line 9: "2.6" instead of 2.5

**Response** :Noted
**Change**: The entire ordering of sect 2 is changed,

Page 17 line 11: delete "from"

**Response** :Noted
**Change**: It is changed, see p.18, l.9 in marked MS

Page 17 line 18: The following procedure was conducted:

**Response** :Noted
**Change**: It is changed, see 19, l.1 in marked MS

Page 18 line 20: delete "for"

**Response** :Noted
**Change**: It is changed, see 20, l.3 in marked MS

Page 19 line 11: delete ")."

**Response** :Noted
**Change**: It is changed, see 21, l.7 in marked MS

Page 20 line 2: What do you mean with "most catchments"? How many catchments have these "snow towers"? Is this phenomenon only observed for high elevated catchments?

**Response**: We agree that the term "most catchments" is not very precise. The high mean annual slope of SWE using SD_LN was the cause of such a statement.

**Change**: In the stratified analysis of the catchments with respect to results SWE and SCA we have included quantification of such behavior and investigated if it is related to mean elevation, catchment size etc. (see response and change to R#1, general comment #4)

Page 20 line 18: You wrote that you found 150 estimates for SCA for each catchment. In page 21 line 4 you wrote that 69 catchments have values for SCA and 2 have no SCA observations. Also why did you write in line 7 70 catchments? Please correct these inconsistencies or explain better!

**Response**: Sorry, a typo. There are 71 catchments. Only 69 catchments have estimated SCA
**Change**: We have changed the numbers, see p.20, l.7-9 in marked MS

Page 21 line 5: delete "for"

**Response** :Noted
**Change**: It is changed, see p.23, l.10 in marked MS

Table 1: On page 16 line 18 you wrote that you use temperature and precipitation lapse rates, but why are they 0 in Table 1? Additionally, I would suggest shortening the table to the most relevant parameters, because you do not use the most of the parameters in the following. Include a space between Table and 1 (page 34 line 1) Also correct "Mean elevation of catchment"

**Response**: They are set to zero since they are not used. Unless the editor wishes otherwise,  we would like to keep the table as it is since it is complete for the DDD model. Just having a subset of the table would demand an additional paragraph explaining the other parameters.
**Change**: We have corrected Table 1 for misspellings, explained about the lapse rates and it has now the format suggested by the Editor, only two columns.  See p.38-40 in marked MS

Table 2: Where does this 1.02 value come from? You wrote in the table caption, that 1 is the ideal value.

**Response**: 1 is indeed the ideal value but the variability error is allowed to be more than 1 (signifies higher variability than the observed series), see Kling et al. (2012), full reference is found in the paper.
**Change**: No change.

Figure 1: "Spatial mean and standard deviation of observed precip." I would additionally suggest including the parameter values of the fitted line and rename "m" on the x-axis to "mean".

**Response** :Noted
**Change**: It is changed accordingly, see p.46 in marked MS

Figure 2: This figure is very hard to understand. Where comes the 0.1 on the x-axis label come from?

**Response**:R#1 had the same comment (specific commet #7). Since we deal with spatial frequency distributions, one must think of the frequencies as number of locations with a given SWE value. The x-axis shows the number of units, so we have to multiply with the unit value (0.1 mm) in order to have mm.

**Change**: We have made a new Figure 2 (see p.47 in marked MS) and elaborated on the explanation, see response to R#1, specific comment #7.

Figure 5: Why do you include a running average over the catchments? Are they sorted by size, mean elevation,..?

**Response**: The running mean was included to improve readability. They are not sorted by size, elevation but geographically, starting with central southern Norway, moving along the coast to the north.
**Change**: An explanation for the moving average is included, see p.21, l.14-15 in marked MS. A new analysis of the results is conducted (see response and change to R#1, general comment #4).

Figure 6: Is your time unit days? It would be better to choose years! What does the "16.75" in the figure caption mean?

**Response**: Yes. "16.75 " is the identification of the catchment"
**Change**: We have added time labels on the x-axis and removed the "16.75". See p.52 in marked MS

Figure 7: I would suggest changing the y limits in the figures a and b to clearer see the differences between the log-normal and gamma distribution. Is the unit of slope of regression "mm" and "C"? I think it should be mm/time and _C/time (_C/year; mm/year)

**Response**: Agreed, to both comments
**Change**: We have changed the figure accordingly, see p.54 in marked MS.

Figure 8: include the unit of the RMSE. Does this mean that the model is around 15% wrong in estimating the SCA? Do the models underestimate or overestimate the SCA? Where are the largest errors observed?

**Response**: We can include the unit and yes, the models are around 15% wrong in estimating SCA.
**Change**: In the more stratified analysis of the results we have answered the questions posed by the reviewer and included units on the y-axis, see p.55 in marked MS.( also see response and change to R#1, general comment #4).

Figure 9: It is very difficult to see anything from this figure.

**Response**: The figure should have proper labels, but we do not see why it is so difficult to read the figure. Red and blue are simulated values of SCA and the green circles represents observed SCA, just as the figure captions says.
**Change**: We have added proper time labels on the axis and included legends, see p.59 in marked MS.

[revised manuscript text omitted]

**Slettet:** As an example,

**Slettet:** distribution of SWE

**Slettet:** In this study we will investigate how snow water equivalent (SWE), snow covered area (SCA) and runoff are simulated when an alternative method for parameterising the spatial distribution of SWE is implemented in a hydrological model. The method has all its parameters estimated prior to calibration and is described in Skaugen (2007) and has since been developed in Skaugen and Randen (2013). The method models the spatial probability density function (PDF) of SWE as a dynamic gamma distribution and is hereafter denoted SD_G (Snow Distribution_Gamma)). SD_G was tested at small test sites and found to model the spatial moments of SWE and SCA well (Skaugen and Randen, 2013), but has, however, not been implemented in a hydrological model and hence not been tested for larger scales and as a tool in operational hydrology.

**Formatert:** Normal

**Slettet:** SCA (Luce and Tarboton, 2004;

**Slettet:** et al., 1999; Liston, 1999; Buttle and McDonnel, 1987). Good simulation of the evolution of SCA is especially important since it controls the runoff dynamics of the spring melt flood and is the basis for properly accounting the energy fluxes in land- surface schemes in atmospheric models (Helbig et al., 2015; Essery and Pomeroy, 2004; Liston, 1999). In addition, remotely sensed SCA is one of the few data measured at the catchment scale for which simulated hydrology can be compared, and represents hence a valuable independent data source to validate models. ¶

**Flyttet ned [1]:** ). The distribution is constant for up to a specified

**Slettet:** (CV)

**Flyttet ned [2]:** and SWE is estimated for nine quantiles and

**Slettet:** In this way, each additional snowfall event has a spatial …

**Flyttet ned [3]:** regardless of its intensity.

**Slettet:** Moreover, the method implies perfect spatial correlation …

[revised manuscript text omitted]

Slettet: northern

Slettet: CV=

Slettet: CV=
Slettet: CV=

Slettet: SD
Slettet: SD
Slettet: From figures 8 and 9 we see that

Naturally, the problem of "snow towers" of DDD_LN influences its ability to simulate a realistic decrease in SCA since small fractions of the catchments remains snow covered at all times. The heavy tails of the optimised accumulation distribution produced by DDD_LN make a complete melt-out of the snow reservoir very difficult. DDD_G, on the other hand, provides an accumulation distribution without the heavy tail, which appears as a better choice with respect to the simulation of both SWE and SCA. The difference between the two methods with respect to the modelling of SCA became very clear when we compared the average annual duration of the snow cover. DDD_LN, due to the positive trends in SWE, ended up with an almost perennial snow cover for most of the catchments (see Figure 9), whereas DDD_G showed a variability in snowcover durations that is more consistent with the varying climate in Norway. For both models the correlation analysis between snow cover duration and CC showed that the duration of snow cover was positively correlated to catchment size, mean elevation and areal fraction of bare rock (area above the treeline) and negatively correlated to the areal fraction of forest. Since the areal fraction of forest and bare rock are highly correlated, these are expected relations illustrating that both models have a realistic snow distribution with respect to elevation.

A more realistically simulated SCA is important for many applications. Updating of snow- and hydrological models using observed SCA is dependent on realistic simulations of SCA. A realistic simulation of SCA is also necessary for the properly accounting of energy fluxes over an area partly covered by snow (Liston, 1999; Essery and Pomeroy, 2004) and is hence important for the assessment of hydrological impacts of climate change. Without realistically simulated SCA, we cannot expect credible simulations for climate projections for neither runoff dynamics nor energy fluxes.

SWE is represented here as the sum of correlated (in time) spatial variables (solid precipitation).

Precipitation events as snow is assumed to be gamma distributed in space with parameters varying with intensity. The parameters, scale, $\alpha_0$, and decorrelation length, $D$, are estimated from observed spatial moments of precipitation. Recall that the shape parameter $\nu_0$, is just set as one tenth of $\alpha_0$ through the relation $E(y) = \frac{\nu_0}{\alpha_0} = 0.1\ mm$. From Figure 1 we see that the variance levels off, and even decreases, for increased spatial mean intensity. The presented model captures this observed feature since the variance will cease to increase as the correlation decreases with intensity (the number of summations). As correlation approaches zero, we will have a sum of independent events. According to the central limit theorem, such a sum will have a normal distribution. The shape parameter of $y$, $\nu_0$ and the correlation determines the rate of the convergence to a normal distribution. For example, if the decorrelation range is long, then more summations are needed for the spatial frequency distribution of SWE to approach a normal distribution. The literature shows that empirical spatial distribution of SWE has a tendency to be positively skewed.  This is especially the case for many observations of SWE in Norway in high alpine areas (Alfnes et al., 2004; Marchand and Killingtveit, 2004; Marchand and Killingtveit, 2005). For more lowland and forested areas, the distribution tend to be more normal (Alfnes et al, 2004; Marchand and Killingtveit,

2004; Marchand and Killingtveit, 2005). In our modelling framework, this would imply that we would expect small shape parameters and long decorrelation lengths for mountain areas, and larger shape parameters together with short decorrelation lengths for lower lying forested areas. Table 4 show correlations and their significance (p-values) between the parameters $\alpha_0$ and $D$ and the CCs fraction of bare rock, fraction of forest, mean elevation and catchment area. We see that $\alpha_0$ is significantly correlated

Slettet:
Slettet: ,
Slettet:  at a certain
Slettet: , and even decreases
Slettet: For uncorrelated events
Slettet: finally
Slettet: .
Slettet: parameter
Slettet:
Slettet: 3
Slettet: with catchment values of
Slettet: ¶

[revised manuscript text omitted]

Slettet: T.
Formatert: Engelsk (Storbritannia)
Slettet: :
Formatert: Engelsk (Storbritannia)
Formatert: Engelsk (Storbritannia)
Slettet: , 2013
Slettet: M. …chirmer, M. and M. …ehning:… M., 2011. Scaling properties and snow depth distribution in an alpine catchment. J. Geophys. Res. 116 D06106, DOI: 10.1029/2010JD014886, 2011
Slettet: J.V. …utcliffe:
Slettet: , 1970
Slettet: ¶ Omhura, A.:
Slettet: , 2001
Slettet: :…, 2007. Uncertainty and multiple objective calibration in regional water balance modelling: a case study in 320 Austrian catchments. Hydrol. Process. 21, 435-446, 200
Slettet: :…, 2004. Estimating fractional snow cover from MODIS using the normalized difference snow index. Remote Sensing of Environment, Vol. 89, pp. 351-361, 2004
Slettet: :
Slettet: , 2012
Slettet: R. …ott, M.…., Lehning, M.…., Schneebeli, M. and
Slettet: :…, 2007. Modelling the spatial variability of snow
Slettet: J. …ndersen:… J., 2010. Simulated precipitation
Slettet: C. …nof (… C., 2014).
Formatert: Engelsk (Storbritannia)
Slettet: I. O.
Formatert
Slettet: A.
Formatert: Engelsk (Storbritannia)
Slettet: Z.
Formatert: Engelsk (Storbritannia)
Slettet: :
Formatert: Engelsk (Storbritannia)
Slettet: 2015.
Slettet: F. …anden… F. 2013. Modeling the spatial
Slettet: :
Slettet: ), 2010.

[revised manuscript text omitted]

**Slettet:** ------------------Sideskift------------------¶
¶

[Figure]

Fig 1

[Figure]

**Slettet:** ----------------------Sideskift----------------------
¶
¶

Fig 2

[Figure]

Fig 3

[Figure]

Fig 4

[Figure]

[Figure]

Fig 5

[Figure]

Fig 6

**Slettet:** ¶
..........................................................Sideskift..........................................................
¶
¶

[Figure]

[Figure]

Fig 7

[Figure]

Fig 8

**Slettet:** ¶
··································Sideskift··································
¶
¶

[Figure]

[Figure]

Fig 9

[Figure]

[Figure]

**Slettet:**

Fig 10

[Figure]

Fig 11

---

## Referee Report (RR1)

The revised manuscript of Skaugen et al. "A model for the spatial distribution of snow water equivalent parameterised from the spatial variability of precipitation" is clearly improved to the former version. The reviewers comments were all answered and considered in the revised manuscript. Readability and clarity is improved. In my opinion the manuscript can be published with some technical corrections.

Details:
The quality of the figures is strongly improved. For me it is still difficult to understand the running mean in Figure 5, 8 and 9, as I expect a running mean for time series but with the explanations in the text and the given regions in the figures acceptable.

There are still some technical corrections to fix e.g. in the references p.32 line 39 or p.33 line 46.

---

## Author Response (AR2)

**Response to re-review of « A model for the spatial distribution of snow water equivalent parameterised from the spatial variability of precipitation" by T. Skaugen and I.H. Weltzien.**

Reviewer 1:
I really appreciate that authors have considered and taken my comments into account. Authors made quite an effort to put the study into more general context and to explain the basic principles of the applied approach. I would recommend publish the paper after considering a few minor comments (mainly improving the clarity of Figure captions):

1) Please consider to indicate in Fig. captions the meaning of S-E, M-N, ..etc.

Response and change: We have revised the figure captions in both Figure 3 and 7 explaining the meaning of the regions.

Maybe these regions can be indicated also in Figure 3, which can be used as a reference for the captions.

Response and change: Figure 3 is changed as suggested by R#1

Please consider to indicate the regions also in Fig.7 (c, d).

Response and change: Figure 7 is changed as suggested by R#1

2) Please check once against the text flow. There are several now well connected "old" a "new" text parts, or duplicates of text (e.g. P.6., l7, P6., l.16, p10 and line 11-16)(pages and lines refer to pdf version with marked changes).

Response and change: We have checked again the flow of the text and revised. In addition to the comments of R#1, we also found some additions typos which we corrected (see marked up MS)

3) It will be interesting to elaborate somewhat more on why do the results sometime differ in different regions of Norway (e.g. why is the DDD_LN better in SW region, or why is there an underestimation of snow?)

Response and change: We have tried to elaborate a bit further in the discussion (see marked up MS, .p.25 and 26)

Reviewer 2:
The revised manuscript of Skaugen et al. "A model for the spatial distribution of snow water equivalent parameterised from the spatial variability of precipitation" is clearly improved to the former version. The reviewers comments were all answered and considered in the revised manuscript. Readability and clarity is improved. In my opinion the manuscript can be published with some technical corrections.

Details:
The quality of the figures is strongly improved. For me it is still difficult to understand the running mean in Figure 5, 8 and 9, as I expect a running mean for time series but with the

explanations in the text and the given regions in the figures acceptable.

There are still some technical corrections to fix e.g. in the references p.32 line 39 or p.33 line 46.
**Response and change: We have corrected the typos in the references.**

[revised manuscript text omitted]

Fig 1

[Figure]

Fig 2

[Figure]

Fig 3

Slettet:

[Figure]

Fig 4

[Figure]

Fig 5

[Figure]

Fig 6

[Figure]

Fig 7

Slettet:

[Figure]

Fig 8

[Figure]

Fig 9

[Figure]

Fig 10

[Figure]

Fig 11